# Heterogenous humoral and cellular immune responses with distinct trajectories post-SARS-CoV-2 infection in a population-based cohort

Dominik Menges [1,8], Kyra D. Zens [1,2,8], Tala Ballouz [1,8], Nicole Caduff[1,2], Daniel Llanas-Cornejo[1], Hélène E. Aschmann[1,3], Anja Domenghino [1,4], Céline Pellaton[5], Matthieu Perreau[5], Craig Fenwick [5], Giuseppe Pantaleo [5], Christian R. Kahlert [6,7], Christian Münz[2], Milo A. Puhan [1] ✉ & Jan S. Fehr[1]

To better understand the development of SARS-CoV-2-specific immunity over time, a detailed evaluation of humoral and cellular responses is required. Here, we characterize anti-Spike (S) IgA and IgG in a representative population-based cohort of 431 SARS-CoV-2-infected individuals up to 217 days after diagnosis, demonstrating that 85% develop and maintain anti-S responses. In a subsample of 64 participants, we further assess anti-Nucleocapsid (N) IgG, neutralizing antibody activity, and T cell responses to Membrane (M), N, and S proteins. In contrast to S-specific antibody responses, anti-N IgG levels decline substantially over time and neutralizing activity toward Delta and Omicron variants is low to non-existent within just weeks of Wildtype SARS-CoV-2 infection. Virus-specific T cells are detectable in most participants, albeit more variable than antibody responses. Cluster analyses of the co-evolution of antibody and T cell responses within individuals identify five distinct trajectories characterized by specific immune patterns and clinical factors. These findings demonstrate the relevant heterogeneity in humoral and cellular immunity to SARS-CoV-2 while also identifying consistent patterns where antibody and T cell responses may work in a compensatory manner to provide protection.

More than two years after its start, the SARS-CoV-2 pandemic remains a threat to public health worldwide and has resulted in hundreds of millions of cases and millions of deaths globally[1]. Several studies have characterized humoral and cellular immune responses against SARS-CoV-2, demonstrating that most people generate both virus-specific antibodies and T cells after infection[2–18]. Virus-specific antibodies have been detected within days of infection[4,13–15,18–20], and their neutralizing capacity has been confirmed[5,11,13,17,20–25], with the magnitude of response

[1]Epidemiology, Biostatistics and Prevention Institute (EBPI), University of Zurich (UZH), Zurich, Switzerland. [2]Institute for Experimental Immunology, University of Zurich (UZH), Zurich, Switzerland. [3]Department of Epidemiology and Biostatistics, University of California San Francisco, San Francisco, CA, USA. [4]Department of Visceral and Transplantation Surgery, University Hospital Zurich (USZ), University of Zurich (UZH), Zurich, Switzerland. [5]Service of Immunology and Allergy, Lausanne University Hospital (CHUV), University of Lausanne (UNIL), Lausanne, Switzerland. [6]Division of Infectious Diseases and Hospital Epidemiology, Cantonal Hospital St. Gallen, St. Gallen, Switzerland. [7]Division of Infectious Diseases and Hospital Epidemiology, Children's Hospital of Eastern Switzerland, St. Gallen, Switzerland. [8]These authors contributed equally: Dominik Menges, Kyra D. Zens, Tala Ballouz. ✉e-mail: miloalan.puhan@uzh.ch

positively correlating with disease severity[6,16,19,26–31]. However, how these antibody responses are maintained longitudinally over time is less clear. While some studies have found stable antibody titers for several months after infection[7,9,23,24,28,31–37], others report substantial declines, particularly among those with mild disease[29,38–41]. The emergence of novel variants of concern and evidence for immune evasion by the Omicron variant[42] has further raised concerns about the longevity of immunity and highlighted the importance of cross-neutralizing antibodies in providing protection from reinfection. In contrast to antibody responses, T cell-mediated immunity may be more stable, and there is evidence that robust T cell responses are developed even after mildly symptomatic coronavirus disease 2019 (COVID-19) or asymptomatic infection and despite low levels or a complete lack of detectable antibodies[7–9,27,32,43,44].

While these studies have significantly advanced the general understanding of immunological responses after SARS-CoV-2 infection, most were conducted in specific populations or using convenience samples. Few have longitudinally assessed multiple components of the immune response within the same individuals[2,7,27,31,32] and in cohorts representative of the full range of the infected population[7,27,31]. Furthermore, despite evidence of heterogeneous immune responses between individuals and the knowledge that antibodies and T cells act together at different stages of SARS-CoV-2 infection to protect against severe disease and reinfection, little attention has been paid to capturing and describing the joint trajectories of antibody and T cell responses within individuals and how these relate to demographic and clinical factors. Since control of the pandemic now relies largely on the development of robust immunity in the population after infection, vaccination, or both, an in-depth understanding of humoral and cellular responses to SARS-CoV-2 is needed.

In this work, we analyze longitudinal patterns of humoral and cellular immune responses for up to 217 days post-diagnosis in a population-based cohort of 431 previously uninfected and vaccine-naïve, wild-type SARS-CoV-2-infected individuals across the COVID-19 disease spectrum. We first characterize anti-Spike (S) IgA and IgG antibody responses and estimate their decay rates. In a subsample of 64 participants selected to cover the clinical spectrum of SARS-CoV-2 infection and S-specific antibody responses, we perform a detailed characterization of anti-Nucleocapsid (N) IgG antibodies, neutralizing antibody activity to Wildtype, Delta (B.1.617.2) and Omicron (B.1.1.529) SARS-CoV-2 variants, and T cell responses specific to viral Membrane (M), N, or S proteins (including the S1 and the majority of the S2 domains). We demonstrate that most Wildtype SARS-CoV-2-infected individuals generate sustained antibody responses which have limited neutralizing activity against Delta and Omicron variants. Furthermore, most individuals generate virus-specific T cell responses, but these are more heterogenous and may mediate virus clearance among those with low antibody responses. Based on the co-evolution of antibody and T cell responses over time, we identify five distinct immune response trajectories which independently correspond to specific patterns of immune responses and clinical features. In summary, we show the heterogeneity of humoral and cellular immune responses following SARS-CoV-2 infection while also identifying consistent patterns where antibody and T cell responses may work in a compensatory manner to provide protection.

## Results
### Study population and sample measurements
In this analysis, we included 431 previously uninfected and vaccine-naïve SARS-CoV-2-infected individuals, selected as a random, age-stratified sample of all SARS-CoV-2-infected individuals reported to the Department of Health of the Canton of Zurich between the 06th of August 2020 and the 19th of January 2021. During this time, nearly all COVID-19 cases were due to infection with the Wildtype virus[45] and vaccines were not yet available. Peripheral blood samples were collected for analysis at two weeks, one month, three months, and six months after diagnosis (Fig. 1a). The median age of participants was 52 years (interquartile range (IQR) 35–68 years), and 212 (49%) were female (Supplementary Table 1). Fifty-nine (14%) of the participants were smokers and 131 (30%) had at least one comorbidity; most commonly hypertension (16%) and respiratory diseases (7%). Most participants (83%) were symptomatic, with 163 (38%) reporting between one and five symptoms and 192 (45%) reporting six or more symptoms during acute infection. Within two weeks of diagnosis, 18 (4%) participants required hospitalization for reasons related to COVID-19, among which two participants were admitted to an intensive care unit. At six months after diagnosis, three participants (0.7%) reported a SARS-CoV-2 reinfection and 80 (19%) had received at least one vaccine dose (all with mRNA vaccines).

We selected a subsample of 64 individuals for a detailed characterization of immune responses, for which we performed additional anti-N IgG antibody testing, neutralization assays, as well as M, N, S1, and S2 domain-specific ELISpot and flow cytometric analyses of virus-specific T cells. The subsample was selected to cover the spectrum of disease severity (i.e., asymptomatic to hospitalized) and antibody responses (i.e., low to high anti-S IgA and IgG antibody responses), while balanced for age and sex (Fig. 1b). The median age of subsample participants was 53.5 (IQR 33.5–68 years) and 56% were female (Supplementary Table 1). 69% of subsample participants had symptoms of COVID-19, with 27% reporting between one and five symptoms and 42% reporting six or more symptoms. Eleven (17%) participants were hospitalized due to COVID-19 and one participant required admission to an intensive care unit.

### Kinetics of S-specific IgA and IgG and N-specific IgG responses
Using a highly sensitive Luminex-based assay[46], we first assessed levels of SARS-CoV-2 S-specific IgA and IgG or N-specific IgG in blood plasma over time. Within the full study population (total $n = 431$), we found that 83% (95% confidence interval (CI) 79–86%) of individuals were anti-S IgA seropositive at two weeks post-diagnosis, decreasing to 70% (65–75%) at six months (Supplementary Fig. 1a, Supplementary Table 2). Median anti-S IgA titers were highest at two weeks post-diagnosis and declined steadily up to six months (Fig. 2a, Supplementary Tables 2 and 3). Similarly, anti-S IgG seropositivity was 82% (78–86%) at two weeks but remained stable at 81% (77–85%) at six months of follow-up (Supplementary Fig. 1a, Supplementary Table 2). Median anti-S IgG titers peaked later, at one-month post-diagnosis, persisted until three months, and then waned slightly until the six-month follow-up (Fig. 2b). Overall, 85% of the population was positive for either anti-S IgA or IgG at three months (81–88%) and six months (81–88%) post-diagnosis. Of note, only ten participants (2.3%) were initially seropositive (either anti-S IgA or IgG) but became seronegative within six months. Estimates were similar when adjusting for the age group-stratified sampling approach (Supplementary Table 4).

Within the subsample (total $n = 64$), we found that 54% (42–66%) were seropositive for anti-N IgG at two weeks, which declined to 41% (29–54%) at 6 months (Supplementary Fig. 1b, Supplementary Table 3). N-specific responses followed a similar initial kinetic to that of anti-S IgG (peaking at one month), but with a more pronounced decline over time (Fig. 2c). Seropositivity for anti-S IgA and anti-S IgG among subsample participants was 59% at two weeks and 61% and 66% at six months, respectively (Supplementary Fig. 1b, Supplementary Table 3). These lower proportions of seropositivity in the subsample were expected, as we aimed to specifically include individuals with low antibody responses in our assessment of T cell responses. To confirm our findings, we further tested these samples using two commercially available assays which detect anti-SARS-CoV-2 S-specific or N-specific total Ig (Roche Elecsys Anti-SARS-CoV-2 S and Roche Elecsys Anti-SARS-CoV-2), respectively. With both additional assays, we observed a high percent agreement (98.7% for anti-S Ig and 91.2% for anti-N Ig

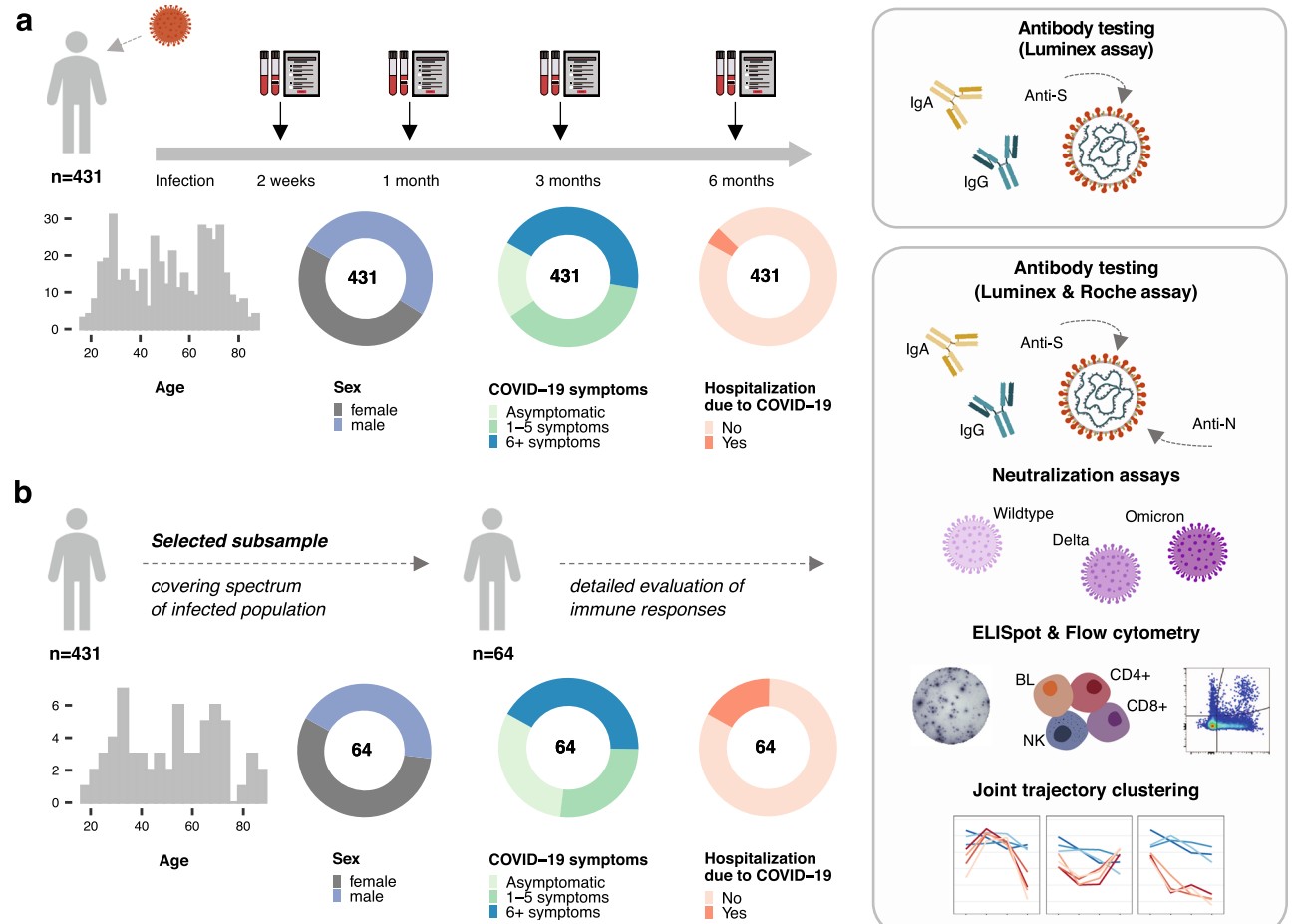

**Fig. 1 | Study design and population characteristics. a** Study design including follow-up timepoints, evaluated immune responses and clinical characteristics of the overall study population (total *n* = 431). **b** Evaluated immune responses and clinical characteristics of the subsample of participants selected to cover the spectrum of the infected population (total *n* = 64). Source data are provided as a Source Data file.

seropositivity; Supplementary Fig. 1c, d) and correlation with the corresponding Luminex-based tests (Spearman *r* = 0.89 for anti-S Ig, Spearman *r* = 0.86 for anti-N Ig; Supplementary Fig. 1e, f), supporting our initial results.

**Rapid loss of neutralizing activity against variants of concern**
To better understand the potential duration of antibody-mediated protection in the light of emerging variants of concern, we evaluated neutralizing activity against Wildtype SARS-CoV-2 and Delta and Omicron variants in the subsample using a cell-free and virus-free surrogate neutralization assay[47]. Less than half (45%, 95% CI 33–58%) of subsample participants (68% of those that were seropositive for at least one evaluated Ig isotype) developed neutralizing responses against Wildtype SARS-CoV-2. These responses peaked at one month post-diagnosis, followed by a rapid decline continuing up to six months (Fig. 2d–f, Supplementary Fig. 2a, b, Supplementary Table 3). Importantly, only 15% (8–26%) of subsample participants (23% of seropositives) developed neutralizing responses against the Delta variant, and only one participant (2%, 0–9%; 3% of seropositives) developed neutralizing responses against the Omicron variant, suggesting only limited protection. Interestingly, neutralizing activity was markedly higher among older participants (Fig. 2d–f).

**Slower anti-S IgG decay vs. anti-S IgA, N IgG, neutralizing activity**
From these data, we next estimated the half-life of each antibody subtype as well as neutralizing antibody activity based on a linear decay model. In the overall study population, we estimated the half-life

to be 71 days (95% CI 66–76 days; Fig. 3a, Supplementary Table 5) for anti-S IgA and 145 days (135–156 days; Fig. 3b) for anti-S IgG. In the subsample, we estimated the half-life of anti-N IgG to be 86 days (76–99 days; Fig. 3c), with half-lives for anti-S IgA and IgG being similar to the overall study population (Supplementary Table 5). For neutralizing antibody responses, we estimated half-lives to be 70 days (55–94 days; Fig. 3d) for anti-Wildtype and 46 days (33–77 days; Fig. 3e) for anti-Delta neutralizing antibodies (anti-Omicron was not estimable). Taken together, these findings suggest that S-specific IgG responses are more robust than S-specific IgA or N-specific IgG responses following SARS-CoV-2 infection. In addition, neutralizing activity declines more rapidly than overall antibody levels and neutralizing activity against Delta and Omicron variants is low to non-existent after a single Wildtype SARS-CoV-2 infection.

**Sustained anti-N, anti-S2 and CD8⁺ T cell responses**
We next evaluated virus-specific T cell populations in the subsample, first using an interferon-gamma ELISpot assay to assess responses to four overlapping peptide pools spanning the entire SARS-CoV-2 M or N proteins, the S1 domain of the S protein (which contains the Receptor Binding Domain (RBD)), or a mix of other predicted immunodominant epitopes within the S protein, containing the majority of the S2 domain (which we refer to here as S2 for simplicity). We found that 84% (95% CI 72–91%) of individuals had detectable T cell responses to at least one or more of the four tested peptide pools at two weeks post-diagnosis, dropping to 71% (58–81%) at six months (Fig. 4a, Supplementary Fig. 2c, Supplementary Table 3). For the pooled T cell response (summed

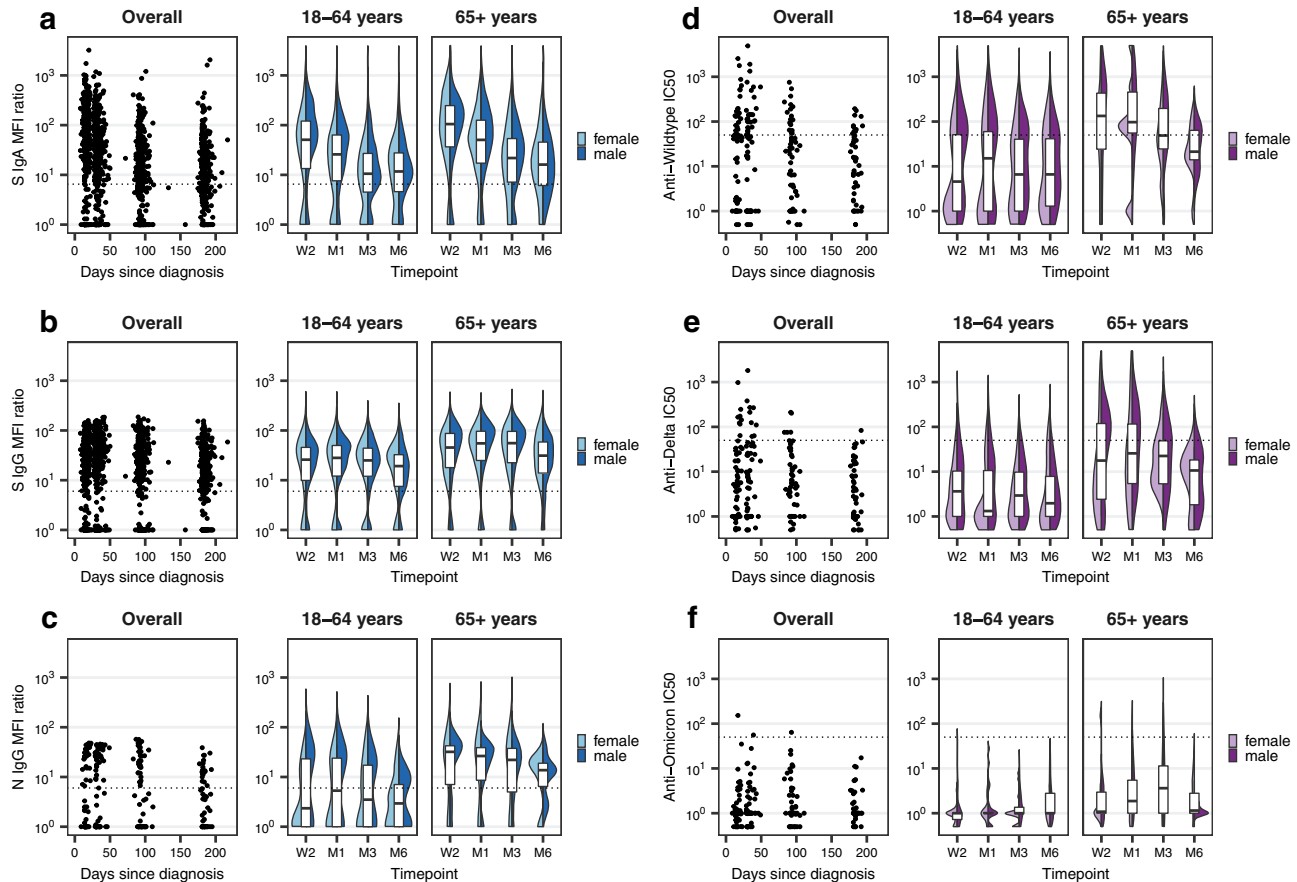

**Fig. 2 | Anti-SARS-CoV-2 S IgA, S IgG, N IgG and neutralizing activity over time.** **a** Anti-S IgA mean fluorescence intensity (MFI) ratios measured by Luminex assay within the overall study population (total $n = 431$; W2: $n = 403$, M1: $n = 421$, M3: $n = 418$, M6: $n = 334$), overall and stratified by age groups and sex. Boxplots in panels **a–f** represent the median and interquartile range (IQR; whiskers: 1.5 × IQR). Dotted lines in panels **a–f** indicate the limit of detection cutoffs (6.5 for IgA MFI ratios, 6.0 for IgG MFI ratios, and 50 for half maximal inhibitory concentrations (IC$_{50}$) for neutralizing activity). W2: two weeks, M1: one month, M3: three months, M6: six months after diagnosis. **b** Anti-S IgG MFI ratios within the overall study

population (total $n = 431$; W2: $n = 403$, M1: $n = 421$, M3: $n = 418$, M6: $n = 334$), overall and stratified by age groups and sex. **c** Anti-N IgG MFI ratios within the subsample of individuals selected for detailed testing (total $n = 64$; W2: $n = 59$, M1: $n = 63$, M3: $n = 64$, M6: $n = 56$). **d** IC$_{50}$ for anti-Wildtype neutralizing antibodies within the subsample (total $n = 64$; W2: $n = 58$, M1: $n = 60$, M3: $n = 58$, M6: $n = 50$). **e** IC$_{50}$ for anti-Delta neutralizing antibodies within the subsample (total $n = 64$; W2: $n = 58$, M1: $n = 60$, M3: $n = 58$, M6: $n = 50$). **f** IC$_{50}$ for anti-Omicron neutralizing antibodies within the subsample (total $n = 64$; W2: $n = 58$, M1: $n = 57$, M3: $n = 49$, M6: $n = 41$). Source data are provided as a Source Data file.

responses to M, N, S1, or S2), we estimated a half-life of 161 days (95% CI 83–2810 days; Supplementary Fig. 2d, Supplementary Table 5).

The proportion of participants positive to individual peptide pools ranged from 71% (95% CI 59–82%; S2) to 80% (68–89%; M) at two weeks to 46% (33–59%; S1) to 51% (38–64%; N) at six months (Fig. 4a, Supplementary Fig. 2c, Supplementary Table 3). The number of detectable M-specific and S1-specific T cells declined significantly over time ($p = 0.03$ and $p = 0.03$, Kruskal–Wallis test; $p = 0.04$ and $p = 0.14$, Friedman test; Fig. 4a), while numbers of detectable N-specific and S2-specific T cells were more stable. Assessing the summed response to all four peptide pools, we found that, at two weeks, M-specific and S1-specific T cells made up 33% and 29%, respectively, of the total detectable T cell response, compared to 26% for each individual antigen at six months (Fig. 4b). In contrast, at two weeks, N-specific and S2-specific T cells made up 21% and 16% of the T cell response, whereas they comprised 27% and 21% of the total response at six months. We further found the half-life of M-specific and S1-specific T cells to be substantially shorter (138 and 137 days, respectively) compared to that of N-specific and S2-specific T cells (251 days and 382 days, respectively), suggesting that the latter may provide more durable responses following infection (Supplementary Fig. 2e–h).

We then assessed virus-specific CD4$^+$ and CD8$^+$ T cell populations in the subsample using a flow cytometry-based Activation-Induced

Marker (AIM) assay. Using the markers CD137 (4-1BB) and CD134 (OX-40) for CD4$^+$ T cells or CD69 and CD137 for CD8$^+$ T cells, we determined frequencies of activated cells following in vitro stimulation with a single, combined megapool of M, N, S1, and S2 peptides (Fig. 4c). We found similar frequencies of total peripheral blood AIM$^+$CD4$^+$ or AIM$^+$ CD8$^+$ T cells (approximately 0.2–0.3% of CD4$^+$ or CD8$^+$) at two weeks post-diagnosis. For CD4$^+$ T cells, this was 0.1% at six months, though this decline did not reach statistical significance ($p = 0.11$, Kruskal–Wallis test; $p = 0.14$, Friedman test; Fig. 4d). In contrast, for CD8$^+$ T cells, frequencies of AIM$^+$ cells remained at 0.2% up to six months post-diagnosis. In assessing the phenotype of AIM$^+$ T cells in the blood, AIM$^+$CD4$^+$ T cells were predominantly central memory T cells (TCM), while AIM$^+$CD8$^+$ T cells were predominantly T effector memory cells re-expressing CD45RA (TEMRA; Fig. 4e). Taken together, our results suggest that anti-S IgG, along with N-specific and S2-specific T cells, may act as more long-lasting components of SARS-CoV-2-specific immunity following infection and that circulating virus-specific CD4$^+$ and CD8$^+$ T cells tend to provide protection through more TCM-like and TEMRA-like functions, respectively.

## Moderate concordance between antibody and T cell responses

Using the data obtained from our analyses of the subsample, we next sought to better understand the relationship within and between

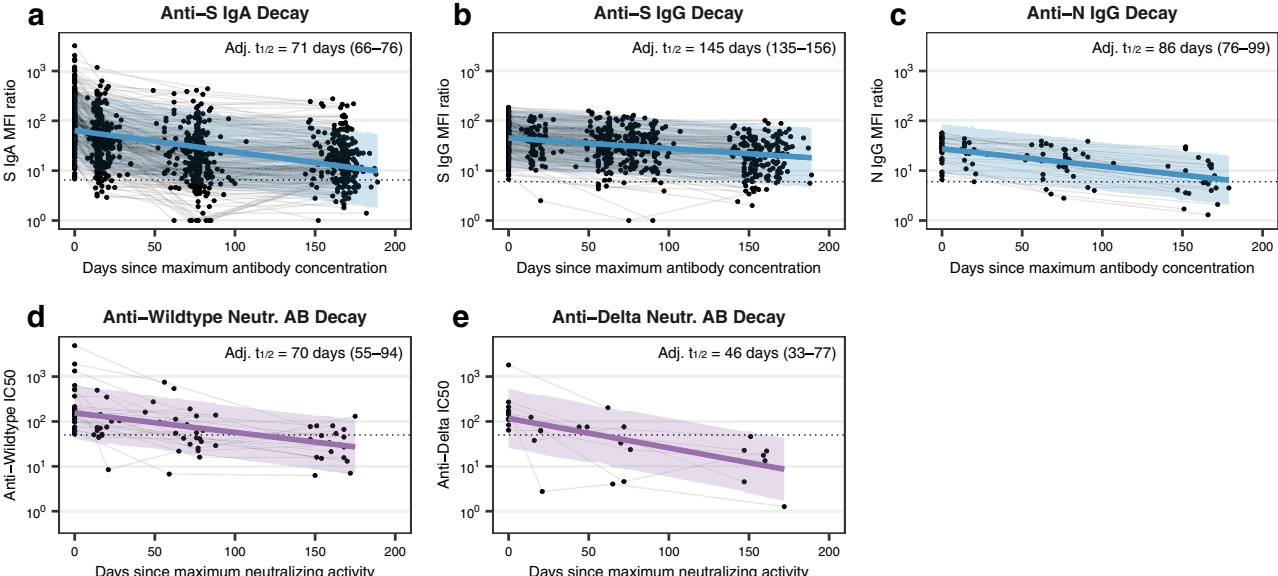

**Fig. 3 | Decay estimation for Anti-SARS-CoV-2 S IgA, S IgG, N IgG and neutralizing activity. a** Anti-S IgA antibody decay estimation within the overall study population based on mixed linear regression model adjusted for time from diagnosis to maximum mean fluorescence intensity (MFI) ratio, age group, sex and symptom count, using a random intercept for individuals. Lines and error bands in panels **a**–**e** represent regression lines with associated 95% confidence intervals estimated using bootstrap. Dotted lines in panels **a**–**e** indicate limit of detection cutoffs (6.5 for IgA MFI ratios, 6.0 for IgG MFI ratios, and 50 for half maximal inhibitory concentrations ($IC_{50}$)). Adj. $t_{1/2}$: adjusted half-life with associated 95% confidence interval. **b** Anti-S IgG antibody decay estimation within the overall study

population based on mixed linear regression model (adjustment as in panel **a**). (**c**) Anti-N IgG antibody decay estimation within the subsample, based on mixed linear regression model (adjustment as in panel **a**). **d** Decay estimation for anti-Wildtype neutralizing antibodies within the subsample, based on mixed linear regression model adjusted for time from diagnosis to maximum $IC_{50}$, age group, sex and symptom count, using a random intercept for individuals. Neutr. AB: neutralizing antibody. **e** Decay estimation for anti-Delta neutralizing antibodies within the subsample, based on mixed linear regression model (adjustment as in panel **d**). Decay estimation for anti-Omicron neutralizing antibodies was not possible. Source data are provided as a Source Data file.

antibody and T cell responses over time. Within the overall antibody response (anti-S IgA, anti-S IgG, anti-N IgG), we observed a strong, positive correlation in the magnitude (assessed as MFI ratios) of responses among different subtypes at each timepoint post-diagnosis (Fig. 5a). Similarly, within the overall T cell response, we observed a strong, positive correlation in the magnitude (assessed as SFU/1e6 PBMCs) of responses among the four different peptide pools (M, N, S1, S2) at each timepoint (Fig. 5a).

In comparing antibody and T cell responses, the correlation was less robust, though still present. At two weeks post-diagnosis, we observed a weak to moderate, positive correlation between the magnitude of each antibody subtype evaluated and pooled T cell (summed M, N, S1, or S2) responses (anti-S IgA, Spearman $r = 0.38$; anti-S IgG, Spearman $r = 0.37$; anti-N IgG, Spearman $r = 0.47$; Fig. 5a). This correlation decreased over time and was weak or no longer observed by six months of follow-up (anti-S IgA, Spearman $r = 0.16$; anti-S IgG, Spearman $r = 0.25$; anti-N IgG, Spearman $r = 0.24$). These findings indicate that antibody and T cell responses tend to behave similarly (in terms of magnitude) early in the immune response but not necessarily in the longer term. Furthermore, they suggest that the magnitude of the response by one antibody subtype is better predicted by the response to other antibody subtypes than by T cell responses and vice versa.

We also evaluated the proportion of individuals that were tested positive (presence of detectable responses) or negative (absence of detectable responses) for antibody and T cell responses at each time point, comparing anti-S IgA, anti-S IgG, or anti-N IgG (MFI ratio above the limit of detection) to overall T cell responses (detectable SFU to one or more peptide pools; Fig. 5b) or to specific, individual peptide pools (Supplementary Fig. 3a). At two weeks post-diagnosis, approximately 55–60% of subsample participants were both antibody positive and overall T cell positive, while approximately 10-15% were both antibody negative and overall T cell negative (Fig. 5b). Thus, the percent

concordance between antibody responses and overall T cell responses was ~70%, and was similar for anti-S IgA, anti-S IgG, and anti-N IgG. By six months, concordance between antibody and overall T cell responses dropped to 55–60% (depending on antibody subtype). For both anti-S IgA and anti-S IgG, this appeared to be primarily due to an increased fraction of participants becoming overall T cell negative. For anti-N IgG, this drop appeared to be additionally driven by an increasing fraction of anti-N IgG negative participants. Patterns of antibody subtype and T cell concordance were also similar between T cells specific to individual peptide pools, and analogous trends were observed when evaluating test agreement using Cohen's Kappa (Supplementary Fig. 3a, b). Together, these findings suggest that SARS-CoV-2-specific antibody and T cell responses correlate early after infection, but that this correlation decreases with increasing time after infection.

**Cluster analysis identifies five distinct joint immune trajectories**
We subsequently explored whether distinct patterns of joint antibody and T cell responses could be observed using a non-parametric longitudinal clustering algorithm[48]. We identified five distinct trajectories of antibody subtype and peptide pool-specific T cell responses within the subsample (Fig. 6a, b). Clusters of participants were primarily defined by the presence (clusters 1-4) or absence (cluster 5) of antibody responses, as well as distinct T cell trajectories. When present, antibody trajectories generally followed the decay patterns observed in the overall study population, characterized by waning anti-S IgA and anti-N IgG and persistent anti-S IgG, though to differing levels. Meanwhile, T cell trajectories between clusters were more heterogenous. We additionally examined neutralization and flow cytometry data, which did not influence clustering, to assess independent differences in immune phenotypes between clusters.

Participants in the first cluster (14% of the subsample, $n = 9$) had the highest initial antibody (anti-S IgA, anti-S IgG, anti-N IgG) and T cell

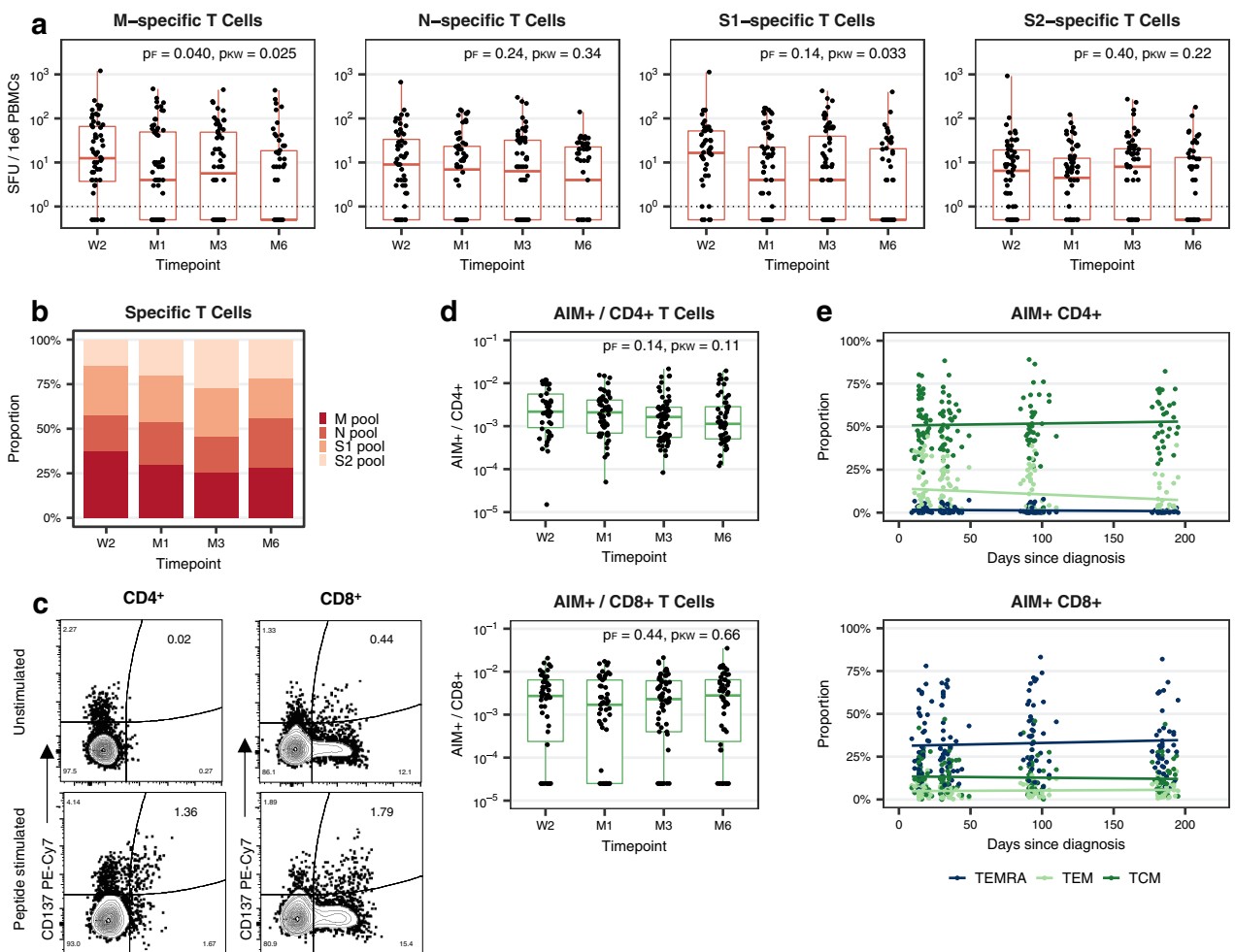

**Fig. 4 | T cell responses specific to SARS-CoV-2 M, N, S1, and S2 epitope pools over time. a** Number of spot-forming units (SFU) per 1e6 peripheral blood mononuclear cells (PBMCs) following stimulation with M, N, S1, or S2 overlapping peptide pools at indicated timepoints post-diagnosis within the subsample of individuals undergoing detailed testing (total $n = 64$; W2: $n = 56$, M1: $n = 64$, M3: $n = 64$, M6: $n = 55$). Boxplots represent the median and interquartile range (IQR; whiskers: $1.5 \times$ IQR). Dotted lines indicate the limit of detection cutoff (SFU values greater than 0). Statistical testing was performed using two-sided Friedman and two-sided Kruskal–Wallis tests to account for both repeated testing and missing data ($n = 48$ with complete follow-up data). $p_F$: $p$-value based on Friedman test, $p_{KW}$: $p$-value based on Kruskal–Wallis test, W2: two weeks, M1: one month, M3: three months, M6: six months after diagnosis. **b** Fraction of the pooled T cell response (summed M, N, S1, and S2 SFU values) specific for each peptide pool at indicated

timepoints post-diagnosis. **c** Representative flow cytometry plots depicting AIM$^+$ (CD134$^+$CD137$^+$) CD4$^+$ (left) and AIM$^+$ (CD69$^+$CD137$^+$) CD8$^+$ (right) T cell populations in unstimulated (top) or SARS-CoV-2 megapool peptide-stimulated (bottom) PBMCs. **d** AIM$^+$ cells as a fraction of CD4$^+$ or CD8$^+$ T cells at indicated timepoints post-diagnosis within the subsample of individuals undergoing detailed testing (total $n = 64$; W2: $n = 46$, M1: $n = 53$, M3: $n = 62$, M6: $n = 54$). Boxplots represent median and IQR (whiskers: $1.5 \times$ IQR). Statistical testing was performed using two-sided Friedman ($n = 35$ with complete follow-up data) and two-sided Kruskal–Wallis tests. **e** Percentage of AIM$^+$CD4$^+$ and AIM$^+$CD8$^+$ T cells with TCM (CD45RA$^-$CCR7$^+$), TEM (CD45RA$^-$CCR7$^-$) or TEMRA (CD45RA$^+$CCR7$^-$) phenotypes by day post-diagnosis. Lines represent trends over time based on unadjusted linear regression models. Source data are provided as a Source Data file.

(M, N, S1, S2) responses, which remained high across all evaluated timepoints (Fig. 6a, b, Supplementary Fig. 4a). Compared to other clusters, we further found that individuals in this cluster had the most robust neutralizing activity against Wildtype SARS-CoV-2 and the Delta variant, and, along with cluster 2, the greatest frequencies of AIM$^+$CD4$^+$ and AIM$^+$CD8$^+$ T cells (Fig. 6c). We noted that participants belonging to the first cluster were mostly older than 65 years (89%), male (56%) and 22% were smokers (Fig. 6d). The majority (78%) had more than six COVID-19 symptoms and 44% were hospitalized within two weeks of diagnosis, indicating more severe disease. Thus, cluster 1 tended to represent older individuals with more severe disease and robust antibody and T cell immune responses.

The second cluster (12% of the subsample, $n = 8$) contained individuals with persistently high anti-S IgA and IgG responses, who also demonstrated a steep increase in virus-specific T cells from two

weeks to one month post-diagnosis (Fig. 6a, b). T cell positivity subsequently declined to nearly non-detectable levels by six months post-diagnosis, with N-specific T cells being the dominant remaining subset, present in 63% of participants (Fig. 6b, Supplementary Fig. 4a). Unlike cluster 1, median neutralizing activity was only above the detection limit for Wildtype SARS-CoV-2 but not the Delta variant. Similar to cluster 1, frequencies of AIM$^+$CD4$^+$ and AIM$^+$CD8$^+$ T cells were higher than in other clusters (Fig. 6c), and most (63%) participants had six or more symptoms and 25% were hospitalized (Fig. 6d). Participants in this cluster were mostly younger than 65 years (63%), female (63%) and non-smokers or ex-smokers (75%). Thus, cluster 2 was composed mostly of younger females with more severe disease, robust antibody responses with limited neutralizing activity, and T cell responses which peaked somewhat later and were mostly undetectable by six months.

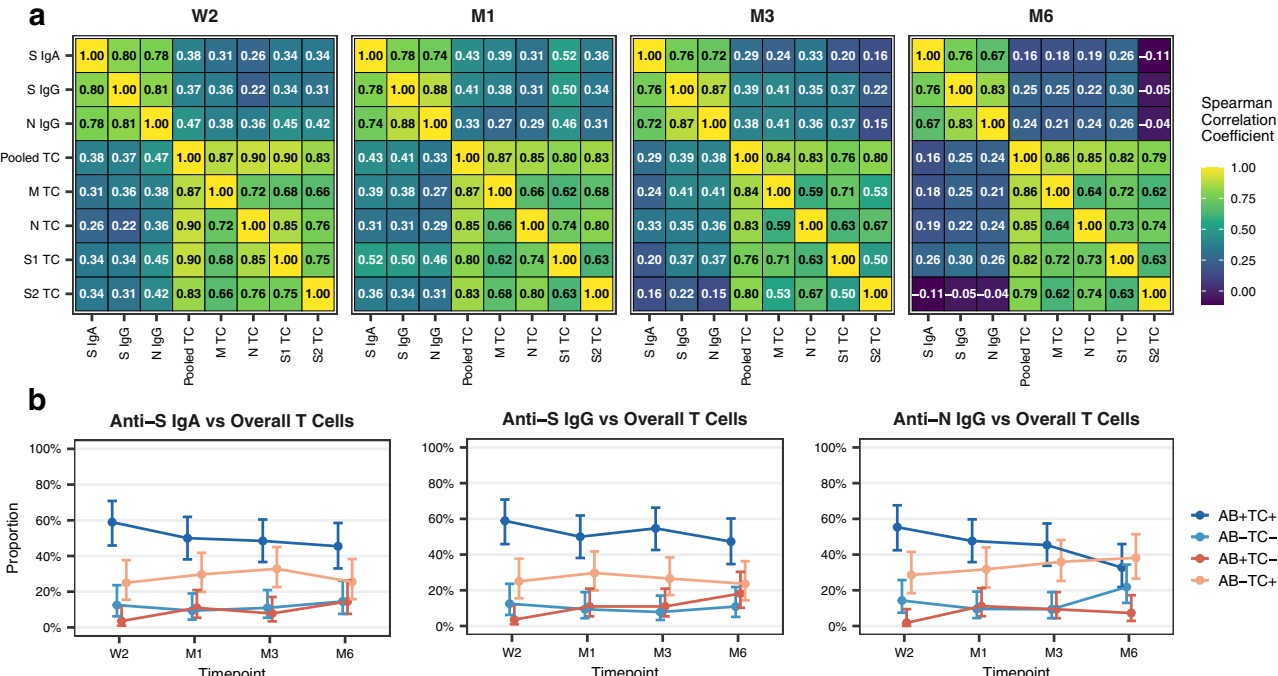

**Fig. 5 | Relationship between antibody and T cell responses over time.**
**a** Heatmaps demonstrating the correlation between anti-S IgA, anti-S IgG, and anti-N IgG levels (as mean fluorescence intensity (MFI) ratios) and M, N, S1, and S2 epitope pool-specific T cells (as spot-forming units (SFU) per 1e6 peripheral blood mononuclear cells (PBMCs)) or pooled T cells (summed M, N, S1, and S2 values) at indicated timepoints. Numbers in individual cells correspond to respective Spearman correlation coefficients. TC: T cell, W2: two weeks, M1: one month, M3: three months, M6: six months after diagnosis. **b** Proportion of participants with concordant and discordant results between testing positive (detectable response) or negative (no detectable response) for anti-S IgA, anti-S IgG, and anti-N IgG antibody subtypes (i.e., MFI ratio values above or below the limits of detection of 6.5 for IgA and 6.0 for IgG) and between being positive or negative for overall T cell responses (i.e., detectable SFU to at least one peptide pool) over time. Points and error bars represent estimated proportions with associated 95% Wilson confidence intervals. AB: Antibody. Source data are provided as a Source Data file.

For individuals in the third cluster (19% of the subsample, $n = 12$), peak antibody responses were observed at two weeks and then declined sharply. Median anti-N IgG responses were below the detection limit by six months while anti-S IgA and IgG were still detectable in most participants (Fig. 6a, b, Supplementary Fig. 4a). Neutralizing responses to both Wildtype and Delta SARS-CoV-2 were generally below the limit of detection (Fig. 6c). T cell responses initially waned between two weeks and one month and then increased again for all peptide pools by three and six months post-diagnosis, at which point 92% of participants had detectable N-specific and S2-specific T cells (Fig. 6a, b, Supplementary Fig. 4a). This pattern was also observed with AIM+CD8+ (but not AIM+CD4+) T cells (Fig. 6c). The characteristics of the participants in this cluster were similar to those in the second cluster, but with only 8% requiring hospitalization during acute infection (Fig. 6d). Together, cluster 3 contains individuals with more mild disease and more moderate, less-well-sustained antibody responses, but with T cell responses which increase over time.

The fourth cluster (16% of the subsample, $n = 10$) was characterized by the presence of antibodies and a rapid decline of T cells specific to all peptide pools between the two week and one month follow-up visits. By six months post-diagnosis, median T cell responses for all peptide pools were below the limit of detection by ELISpot assay, though AIM+CD8+ T cells could be detected in several participants (Fig. 6b, c, Supplementary Fig. 4a), possibly indicating a T cell differentiation stage with limited interferon-gamma production. This cluster was characterized by participants who were predominantly male (80%), younger than 65 years (70%), and non-smokers or ex-smokers (100%; Fig. 6d). About 40% reported having six symptoms or more, and 40% were hospitalized in the acute phase. Together, cluster 4 appears to be comprised of young males with moderate to severe disease but who also have less-well-sustained antibody and T cell responses.

Participants in the fifth cluster (39% of the subsample, $n = 25$) demonstrated persistently low and primarily negative antibody responses with variable T cell responses. Half of the participants had detectable M-specific and N-specific T cells across all assessments (Supplementary Fig. 4a). In general, however, median T cells responses tended to be lower than for other clusters by ELISpot and by AIM assay. Most individuals within this cluster were younger than 65 years (80%) and female (68%). Most participants also had mild disease as reflected by the low reported symptom count, with none of the individuals requiring hospitalization (Fig. 6d). That individuals in cluster 5 experience relatively mild disease in the absence of substantial antibody responses suggests compensatory protection through cellular immune responses, possibly in addition to the monitored T cell responses including T cells localized to tissues or specific to viral proteins not captured by the assays used in this study.

Overall, our findings highlight the substantial heterogeneity in antibody and circulating T cell responses between individuals after SARS-CoV-2 infection while also demonstrating the presence of distinct joint immune trajectories within which individuals share similar patterns of immune phenotypes as well as demographic and clinical characteristics.

## Demographic and clinical factors associated with immune responses
Lastly, we evaluated whether individual demographic and clinical factors described within the clusters were associated with immune responses in the overall study population using adjusted mixed-effects linear regression analyses. Consistent with other reports[6,16,19,26–29,31,49], we found that older age (≥65 years; $p < 0.001$, $t$-test based on mixed-effects

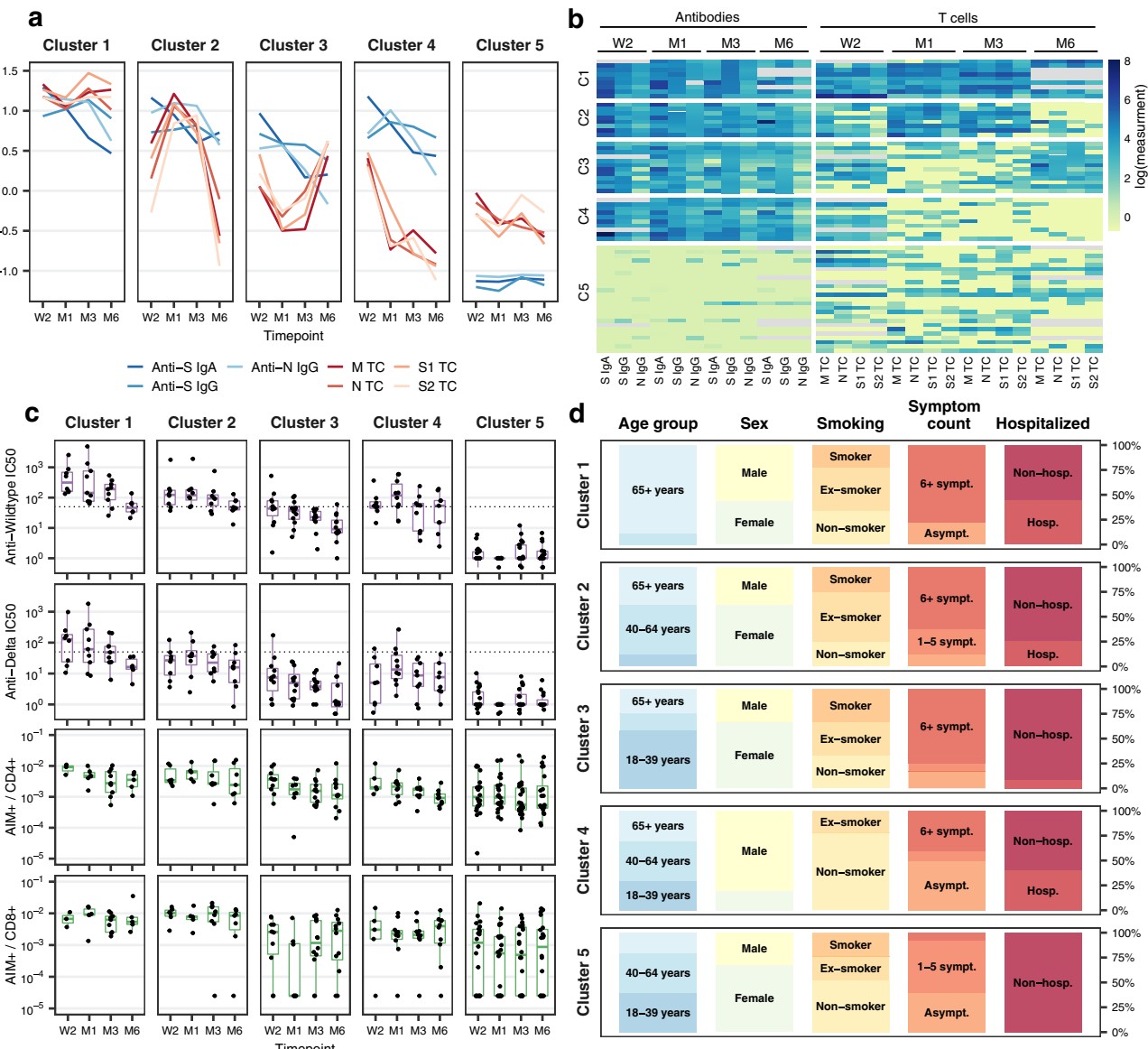

**Fig. 6 | Clustering of antibody and T cell response trajectories. a** Average mean fluorescence intensity (MFI) ratios of anti-S IgA, anti-S IgG, and anti-N IgG antibodies and mean M, N, S1, and S2 epitope pool-specific T cells (spot-forming units (SFU) per 1e6 peripheral blood mononuclear cells (PBMCs)) in each of the five clusters over time (total $n = 64$; cluster 1: $n = 9$, cluster 2: $n = 8$, cluster 3: $n = 12$, cluster 4: $n = 10$, cluster 5: $n = 25$). Displayed data was natural logarithm-transformed and normalized (rescaled). W2: two weeks, M1: one month, M3: three months, M6: six months after diagnosis. **b** Heatmap showing anti-S IgA, anti-S IgG, and anti-N IgG MFI ratios and M, N, S1, and S2 epitope pool-specific SFU/1e6 PBMCs for all participants belonging to the five identified clusters. Gray color indicates missing values, displayed data corresponds to natural logarithm-transformed measured data. C1–5: cluster 1 to 5.

**c** Anti-Wildtype SARS-CoV-2 neutralizing antibody half maximal inhibitory concentration ($IC_{50}$), anti-Delta SARS-CoV-2 neutralizing antibody $IC_{50}$, frequency of $AIM^+CD4^+$ and $AIM^+CD8^+$ T cells in the five clusters over time. Boxplots represent the median and interquartile range (IQR; whiskers: 1.5*IQR). Dotted lines indicate limit of detection cutoffs (50 for $IC_{50}$ for neutralizing activity). **d** Distribution of participant characteristics within the five clusters according to age group (18–39 years, 40–64 years, ≥65 years), sex (male and female), smoking status (non-smoker, ex-smoker, smoker), number of symptoms reported (asymptomatic, 1–5 symptoms, ≥6 symptoms), and hospitalization status (non-hospitalized, hospitalized) during acute infection. Asympt.: asymptomatic, sympt.: symptoms, hosp.: hospitalized. Source data are provided as a Source Data file.

linear regression model using Satterthwaite's method[50]), male sex ($p = 0.011$, $t$-test via Satterthwaite's method), higher symptom severity including one to five ($p < 0.001$, $t$-test via Satterthwaite's method) or more than six COVID-19 symptoms ($p < 0.001$, $t$-test via Satterthwaite's method), as well as hospitalization ($p = 0.006$, $t$-test via Satterthwaite's method) were all independently associated with higher anti-S IgG MFI ratios over time (Fig. 7a, Supplementary Table 6). Conversely, being a current smoker was associated with lower responses ($p = 0.010$, $t$-test via Satterthwaite's method). Results for anti-S IgA MFI ratios in the overall population were comparable (Fig. 7b, Supplementary Table 7), and similar trends were identified for anti-N IgG MFI ratios

within the subsample (Fig. 7c). For T cell responses, we found that older age (≥65 years; $p < 0.001$, $t$-test via Satterthwaite's method) and having more than six symptoms ($p < 0.001$, $t$-test via Satterthwaite's method) during acute infection were each independently associated with higher T cell numbers over time within the subsample (Fig. 7d, Supplementary Table 8). Sensitivity analyses for antibody or overall T cell positivity at two weeks and six months after diagnosis using logistic regression models showed comparable results (Supplementary Tables 9 and 10). Therefore, the severity of disease, age and smoking status are all independently associated with the magnitude of immune responses.

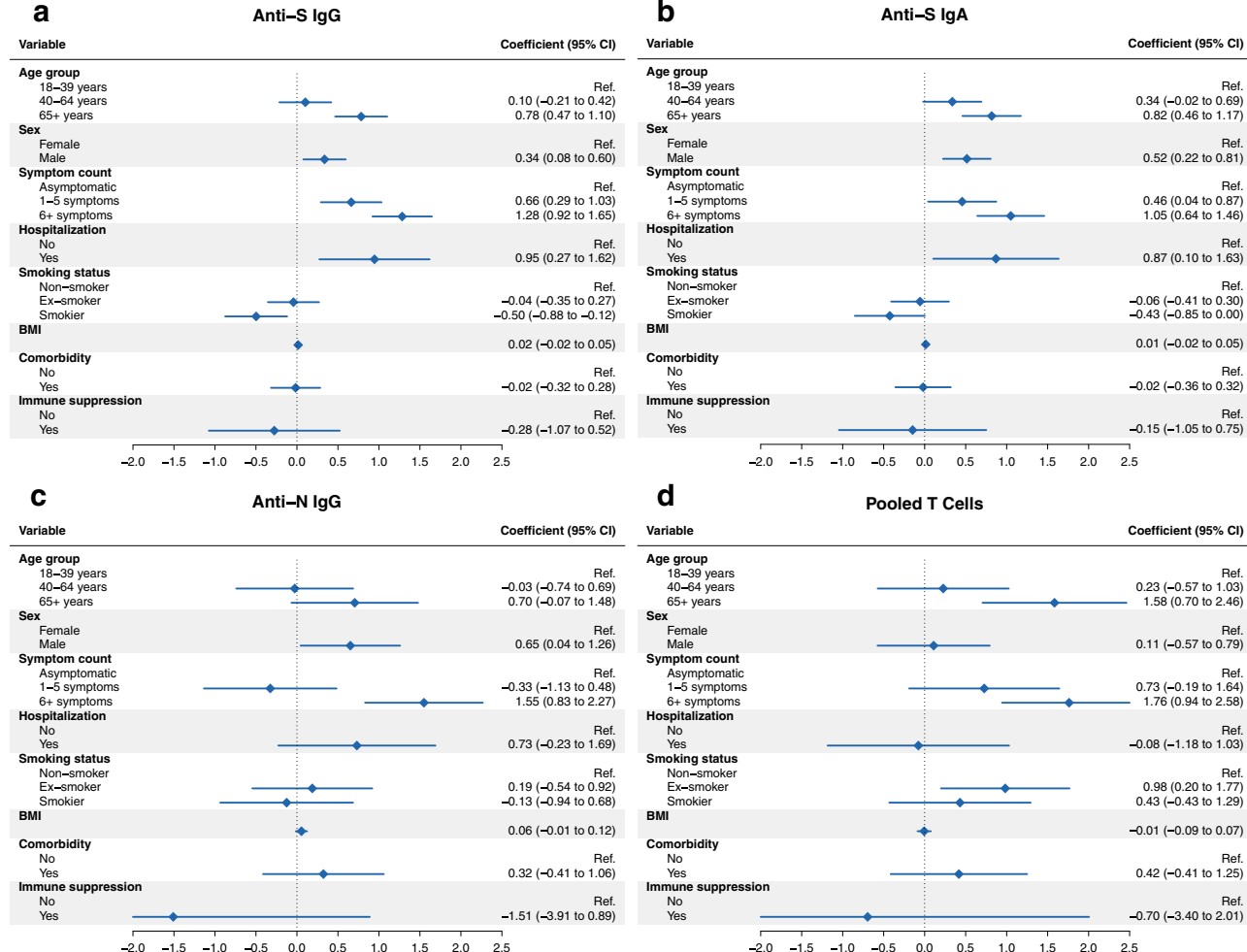

**Fig. 7 | Association analyses of antibody and T cell responses. a** Forest plot demonstrating results from adjusted mixed linear regression analyses evaluating the associations of anti-S IgG mean fluorescence intensity (MFI) ratios with demographic and clinical factors (age, sex, COVID-19 symptom count, smoking status, body mass index, presence of at least one comorbidity and immunosuppression) in the overall study population (total *n* = 431). The model was adjusted for time since diagnosis, age group, sex, and disease severity expressed as symptom count, with a random intercept for each individual. Points and error bars in panels **a**–**d** represent estimated coefficients with associated 95% confidence intervals (CIs). BMI: body mass index. **b** Forest plot showing results from adjusted mixed

linear regression analyses evaluating the associations of anti-S IgA MFI ratios with demographic and clinical factors in the overall study population. Model adjustment as for panel **a**. **c** Forest plot showing results from adjusted mixed linear regression analyses evaluating the associations of anti-N IgG MFI ratios with demographic and clinical factors in the subsample (total *n* = 64). Model adjustment as for panel **a**. **d** Forest plot demonstrating results from adjusted mixed linear regression analyses evaluating the associations of pooled T cells (summed M, N, S1, and S2 spot-forming units (SFU) per 1e6 peripheral blood mononuclear cells (PBMCs)) with demographic and clinical factors in the subsample. Model adjustment as for panel **a**. Source data are provided as a Source Data file.

## Discussion

Further understanding of the characteristics and trajectories of immune responses after SARS-CoV-2 infection remains important with the rollout of vaccines and the emergence of novel variants of concern worldwide. Here, we provide a longitudinal evaluation of simultaneous humoral and cellular immune responses from shortly after and up to 217 days after SARS-CoV-2 infection in a sample of recovered, previously uninfected and vaccine-naïve individuals covering the spectrum of clinical disease and S-specific antibody responses.

We demonstrate that approximately 85% of individuals infected with SARS-CoV-2 developed detectable circulating S-specific IgA and IgG antibody responses and that these responses are maintained for up to six months. Compared to S-specific IgG, the proportion of participants with detectable anti-S IgA decreased markedly over time (from 85% to 70% at six months). Consistent with this, we found that the half-life of anti-S IgA was considerably shorter (71 days, with a range of 42 to 210 days reported by other studies[2,7,17]) compared to that of anti-S IgG (145 days, with a range of 36 to 245 days reported by other

studies[2,7,10,34,51–54]). This difference, however, is perhaps not surprising as the half-life of IgA, in general, is shorter than that of IgG. We further found that anti-N IgG waned more rapidly than anti-S IgG (estimated half-life 86 days). These findings are consistent with other work demonstrating the importance of anti-S IgG, in particular, in sustained protection against SARS-CoV-2. Furthermore, they indicate that both the target of the antibody response as well as the isotype itself appear to influence the duration of humoral immune responses after infection. We cannot rule out that some individuals may only have very low levels or different specificities of circulating antibodies or that they have mucosal, but not circulating, antibody responses which we would not be able to detect here. However, we were able to validate our findings by utilizing two distinct and well-established[46] serology assays.

With regard to neutralizing antibody responses, we found that less than half of subsample participants demonstrated neutralizing activity against Wildtype SARS-CoV-2. Neutralizing activity against the Delta variant was low and almost non-existent against the Omicron

variant. While our half-life estimates (70 days for anti-Wildtype and 46 days for anti-Delta neutralizing antibodies) are longer than those reported by others[7,25], they still clearly indicate that neutralizing activity wanes more rapidly than overall antibody responses.

In the subsample, we also found that the majority had detectable, interferon-gamma-producing T cell responses to at least one of the four SARS-CoV-2-specific peptide pools analyzed in this study. Based on this polyclonal T cell response, we estimated a half-life of 161 days, consistent with other reports[2,7]. Overall T cell positivity (a detectable response to at least one peptide pool) in the subsample (nearly 85%) was higher than for overall antibody seropositivity (just over 60%), suggesting that T cell responses are likely able to provide protection even in the absence of antibodies. However, as only a limited range of antibody and T cell responses were tested in this study, it could also be possible that mucosal, rather than circulating responses, or responses to other viral proteins other than those analyzed here, are capable of providing this protection.

Interestingly, the T cell peptide pools for which individuals tested positive were heterogenous. For example, at six months, 50% of individuals were positive for any given individual pool despite overall T cell positivity of approximately 70%, which illustrates the polyclonal nature of the anti-viral T cell response. While M and S1 appeared to be immunodominant in terms of the magnitude of T cell responses, they also decayed more rapidly, with half-lives of 138 and 137 days, compared to N and S2-specific responses with half-lives of 251 and 382 days, respectively. Compared to S1, the S2 domain of the full-length spike protein shares a higher degree of amino acid identity with endemic coronaviruses[55]. For example, HCoV-HKU1 shares an amino acid identity of 42% with SARS-CoV-2 at the S2 domain of the spike protein compared to 31% at the S1 domain. Therefore, the longer half-life of S2-specific compared to S1-specific T cells, could potentially be due to intermittent exposure to endemic coronaviruses with similar S2 peptide sequences, resulting in the appearance of a prolonged and more durable T cell response. Overall, our findings suggest that M and S1 responses may initially be more robust, at least in terms of an interferon-gamma-producing response, but that perhaps responses to N and S2 may be more durable.

In assessing virus-specific CD4⁺ and CD8⁺ T cells by AIM assay, we found that frequencies of AIM⁺CD4⁺ or AIM⁺CD8⁺ T cells were similar at two weeks post-infection, but that activated, virus-specific CD4⁺ T cells appeared to decline slightly over the six months of follow-up compared to AIM⁺CD8⁺ T cells. Furthermore, AIM⁺CD4⁺ T cells were predominantly of TCM phenotype, and this was consistent throughout the convalescent period, whereas AIM⁺CD8⁺ T cells had a predominantly TEMRA phenotype, in agreement with the previous studies[7,56,57]. These findings support the idea that virus-specific CD4⁺ T cells may contribute to maintaining the immune response and immunological memory, whereas CD8⁺ T cells may more directly contribute to the anti-viral immune response as highly activated and more terminally-differentiated cytotoxic lymphocytes.

Regarding the relationship between antibody and T cell responses within individuals over time, as well as the patterns of these responses within the population, we found a strong, positive correlation within the three antibody subtypes evaluated here, as well as within T cells specific to M, N, S1, and S2. This suggests that, while there may be some variation between distinct subtypes or specificities, both overall antibody and T cell responses tended to behave similarly. The relationship between antibody and T cell responses, however, was less strong, with a weak to moderate positive correlation early after infection, which decreased over time. Consistent with this, more than 70% of participants had concordant results at two weeks (58% both antibody and T cell positive and 13% both antibody and T cell negative) compared to 55–60% at six months. These findings suggest that individuals have heterogeneous immune responses following infection (possibly due to differences in viral load or

primary site of infection or previous immune history) and that they perhaps retain different subsets of immune memory components which could be recalled upon reinfection.

Based on this idea, we explored whether there were potential patterns of immune trajectories which individuals tend to follow in response to infection and which might influence not only their response to infection but also the immune memory populations which they establish. Using a longitudinal clustering algorithm, we identified five distinct joint trajectories of antibody and T cell responses within the subsample of individuals selected to cover the spectrum of clinical disease and S-specific antibody responses. We observed that clusters (each containing between eight and 25 individuals) independently demonstrated distinct patterns of neutralization activity and AIM⁺CD4⁺and AIM⁺CD8⁺ T cell responses and that there were distinct clinical characteristics among individuals in each cluster. Individuals in clusters 1 and 2 had the most robust immune responses and also tended to be older and to have had more severe COVID-19, as reflected by hospitalization and the number of reported symptoms. Older age and disease severity have been reported by several other studies to be associated with higher immune responses to SARS-CoV-2[6,16,19,26–29], and similar observations were made in patients with severe COVID-19 or requiring hospitalization[4,15,19]. Using statistical modeling, we were able to confirm independent associations between older age, male sex and higher disease severity with stronger immune responses in the overall study population.

Compared to clusters 1 and 2, antibody and T cell responses in clusters 4 and 5 appeared to be less robust. Cluster 5 consisted mainly of younger females who reported mild COVID-19 or asymptomatic infection. While we did not detect antibody responses in these individuals, T cell responses were present in half, which, in conjunction with the generally less severe presentation, may suggest some form of compensatory T cell-mediated protection from severe COVID-19 and viral clearance by T cells. In cluster 3, we noted a considerable increase in virus-specific T cells, and, to a lesser extent, in anti-S IgA, after an initial decline between two weeks and one month after infection. In combination with the increase in N-specific T cells, we hypothesized that this could be due to viral re-exposure. Of note, about half of cluster participants underwent SARS-CoV-2 testing between three and six months, suggestive of exposure events or symptomatic episodes. None, however, reported testing positive. It is also possible that, rather than re-expansion, the decrease at one to three months reflects the trafficking of lymphocytes from the circulation into the tissues. Indeed, levels of total CD4⁺ T cells, B cells and NK cells also tended to be lower in this cluster at these timepoints (Supplementary Fig. 4b). Studies of lung lavage and tissue samples from COVID-19 patients have demonstrated expression of tissue homing and tissue residence marker expression on T cells[58–61]. As we were only able to obtain blood from participants, we were unable to address the issue of site-specific immunity in the current study, though a better understanding of the spatial nature of the immune response to infection is certainly necessary.

Our cohort is one of few population-based and longitudinal studies providing a detailed assessment of antibody and T cell responses in a sample of individuals representative of the spectrum of SARS-CoV-2 infection, including asymptomatic to severe disease courses. However, some limitations should be considered when interpreting our findings. First, we captured relatively few critically-ill patients in our study, which may show distinct immune response patterns[62–64]. Second, we primarily relied on a Luminex-based assay to quantify antibody responses in our study and test accuracy may have influenced our results. However, the Luminex assay has been extensively validated prior to this study (Supplementary Fig. 1f) and was shown to be highly sensitive and specific[46]. In addition, we performed a full validation within the subsample participants using two commercially available assays, which showed high test agreement and confirmed the

Luminex results. Third, we used a surrogate assay to indirectly quantify neutralizing activity by measuring the competitive inhibition of tri-meric SARS-CoV-2 S protein binding to the Angiotensin Converting Enzyme 2 (ACE2) receptor. However, this high-throughput assay showed very high sensitivity compared to live virus assays during validation[47] and enabled the simultaneous assessment of neutralization against Wildtype SARS-CoV-2, Delta and Omicron variants. Fourth, we limited our T cell analysis to a single ELISpot assay for the three dominant antigens for cellular immune responses (S, M and N)[2,7,65]. We cannot exclude the importance of subdominant T cell responses against other viral antigens in some of the participants, which may have led to an underestimation of the proportion of individuals with T cell responses. Fifth, the cluster analysis bears the limitations that are inherent to the methodology. Such algorithms may not necessarily result in clusters that are reflective of clinically meaningful differences. However, we found distinct clinical and immunological correlates within the identified clusters, including various measurements (such as data from neutralization assays and flow cytometric analyses) that were not used within the clustering model. Therefore, we consider our results to be relatively robust and leading to a meaningful description of different immune trajectories. Furthermore, the limited sample size of the subsample was a pragmatic choice to ensure the feasibility of the project. It cannot be excluded that further relevant immune response patterns may have been observable with additional data. Finally, we analyzed antibody and T cell testing results for only up to six months, limiting our findings regarding the long-term durability of immune responses to SARS-CoV-2. Nonetheless, this study provides a unique in-depth analysis of joint humoral and cellular immune response trajectories, which may lead to further insights on the variability of SARS-CoV-2-related immune responses and may also be relevant regarding emerging variants of concerns or potential future pandemics.

In this study, we provide important insights into the dynamics and heterogeneity of antibody and T cell immune responses among SARS-CoV-2-infected individuals. We identified five distinct immune trajectory patterns which were representative of clusters of individuals with distinct immune features and demographic and clinical characteristics. While antibody and T cell responses strongly correlate in some individuals, their discordance in others highlights the complex interactions of the immune system among infected individuals and indicates that there are several mechanisms by which protection against SARS-CoV-2 infection can be achieved.

## Methods

### Study design and participants
We recruited a population-based, age-stratified, random sample of 431 individuals diagnosed with SARS-CoV-2 infection between the 6th of August 2020 and the 19th of January 2021 in the Canton of Zurich, Switzerland (Supplementary Fig. 5). The study protocol was approved by the Cantonal Ethics Committee of Zurich (BASEC Registration No. 2020-01739) and prospectively registered (ISRCTN 14990068)[66].

Study participants were identified through the Department of Health of the Canton of Zurich, which records all diagnosed SARS-CoV-2 cases within the Canton through mandatory case reporting. Eligibility criteria were having a polymerase chain reaction (PCR)-confirmed SARS-CoV-2 diagnosis, being aged 18 years or older, residing in the Canton of Zurich, understanding the German language, and being cognitively able to follow the study procedures. We obtained written informed consent from all participants upon study enrollment. Participants were compensated with a flat fee for any travel expenses related to study visits but otherwise did not receive any compensation for their participation.

We collected peripheral venous blood samples during study visits at two weeks, one month, three months and six months post-diagnosis. Participants additionally provided information regarding acute COVID-19 disease course, severity and symptoms, longer-term health and complications, past medical history, and socio-demographics at the corresponding timepoints through electronic questionnaires. We used the Research Electronic Data Capture (REDCap) platform for data collection[67]. Median follow-up was 183 days (range 13–217 days, inter-quartile range (IQR) 181–186 days), with five participants lost to follow-up (Supplementary Fig. 5).

We selected 64 out of the 431 participants for a detailed char-acterization of immune responses. This sample was aimed to be representative of the spectrum of SARS-CoV-2 infection and associated immune responses. Participants in this subsample were selected at random within strata based on clinical characteristics (asymptomatic disease, low and high symptom count, hospitalization) and S-specific antibody responses up to one month (seronegative or low anti-S IgA or IgG response, seropositive or high anti-S IgA or IgG response), while ensuring balance across sex and age groups. Based on preliminary assessments[46,47] and evidence from other studies[4,8,12,15,19,26,44], we deemed this sample size to be a sensible and pragmatic choice allowing us to identify the range of immune responses in infected individuals, while ensuring the feasibility of the project.

### Isolation of plasma and PBMCs
Blood samples collected from participants in K2-EDTA vacutainer tubes (BD) at each study timepoint were subjected to initial centrifugation to collect plasma, followed by isolation of peripheral blood monocytic cells (PBMCs) from the remaining cellular fraction by density-gradient centrifugation using Ficoll–Paque (density 1.077 g/ml). Plasma aliquots were stored at −20 °C prior to IgA and IgG antibody analyses. PBMCs were initially frozen in 90% fetal bovine serum (FBS, Pan Biotech) with 10% dimethyl sulfoxide (DMSO, Sigma) at −80 °C and transferred to liquid nitrogen prior to use in ELISpot and Activation Induced Marker (AIM) flow cytometry assays.

### Analysis of Spike-specific IgA and IgG and Nucleocapsid-specific IgG
Cryopreserved plasma samples were thawed and analyzed for levels of Spike (S)-specific IgA and IgG, or Nucleocapsid (N)-specific IgG by Luminex assay[46]. In brief, assay beads were prepared by covalent coupling of either the SARS-CoV-2 Spike protein trimer or Nucleo-capsid protein with MagPlex beads using a Bio-Plex 356 Amine Cou-pling Kit (Bio-Rad, Catalog 10000148774) per manufacturer's protocol. Protein-coupled beads were diluted and added to each well of Bio-Plex Pro 96-well Flat Bottom Plates (Bio-Rad). Beads were washed with PBS on a magnetic plate washer (MAG2x program), and 50 µl of individual plasma samples diluted 1:300 in PBS were added to plate wells. A pool of pre-COVID-19 pandemic healthy human sera was used as a negative control (BioWest human serum AB males; VWR). Plates were incubated for 1 hour at room temperature with shaking, washed with PBS and incubated with 50µl of a 1:100 dilution of poly-clonal Goat F(ab')2 anti-human IgA-PE (for anti-IgA assay; Southern Biotech; Catalog 2052-09) or polyclonal Goat anti-human IgG-PE (for anti-IgG assay; OneLambda, Catalog LS-AB2) secondary antibody at room temperature for an additional 45 minutes with shaking. After incubation, samples were washed with PBS and resuspended in read-ing the buffer and read on a Bio-Plex (Luminex) 200 plate reader with Bio-Plex Manager software (version 6.2; Bio-Rad) to obtain a mean fluorescence intensity (MFI) value for each sample. The MFI value for each plasma sample was divided by the mean value of the negative control samples to yield an MFI ratio. Based on negative control samples and samples from PCR-positive donors, seropositivity was determined based on MFI ratio cutoff values exceeding 6.5 for IgA and 6.0 for IgG[46]. For this study, the lower limit of measured MFI ratios was restricted to 1, representing equivalent fluorescence intensity com-pared to negative control samples.

Luminex assay results were further validated by two commercial assays designed to detect anti-SARS-CoV-2 S-specific or N-specific total

Ig (Elecsys Anti-SARS-CoV-2 S, Roche, Catalog 09289267190 and Elecsys Anti-SARS-CoV-2, Roche, Catalog 09203095190), respectively, using a Cobas e411 analyzer instrument (software version 03-02; Roche). Plasma samples from subsample participants were assayed according to manufacturer's instructions. In brief, 20 µl of participant samples were incubated with a mixture of biotinylated and ruthenylated RBD antigen (for the S-assay) or a mixture of biotinylated and ruthenylated nucleocapsid antigen (for the N-assay) to form double-antigen-sandwich immune complexes. Afterward, streptavidin-coated magnetic microparticles were added. Within the measuring cell of the instrument, streptavidin microparticle-double-antigen-sandwich complexes were magnetically captured and washed. Voltage was applied to induce electrochemiluminescence (ECL) which was measured with a photomultiplier within the instrument, where increased ECL signals correspond to increased antibody titers. For the Roche Elecsys anti-S Ig assay, values were expressed as U/ml. For the Roche Elecsys anti-N Ig assay, values were expressed according to a cutoff index (COI). Seropositivity was determined based on cutoff values exceeding of a concentration of >0.8 U/ml for anti-S Ig and a COI value of 1.0 for anti-N Ig assays.

In addition, external cross-validations of the Luminex anti-S IgG assay with the Roche Elecsys anti-SARS-CoV-2 S Ig assay were performed prior to this study based on 900 samples from SARS-CoV-2-infected individuals of the Lausanne University Hospital, Switzerland (individuals were not part of the reported population-based study; results for 298 samples reported in Supplementary Fig. 1f). Based on this data, an approximate conversion for anti-S IgG MFI ratios obtained by Luminex assay to anti-S Ig U/ml obtained using the Roche Elecsys Anti-SARS-CoV-2 S assay was determined using a linear regression model of log10-transformed data (Supplementary Table 11).

### Neutralization assays

Cryopreserved plasma samples were thawed and evaluated for the presence of SARS-CoV-2 neutralizing antibodies using a cell-free and virus-free assay[47]. Briefly, 50 µl of diluted plasma samples (1:10, 1:30, 1:90, 1:270, 1:810, and 1:2430) were incubated with Luminex beads covalently coupled to the original SARS-C2oV-2 Spike protein (2019nCoV) and Spike variants Delta (B.1.617.2) and Omicron (B.1.1.529) in Bio-Plex Pro 96-well Flat Bottom Plates (Bio-Rad) for 60 min at room temperature with shaking. Negative control wells on each plate included beads alone and dilutions of pooled, pre-COVID-19 pandemic healthy human sera (BioWest human serum AB males; VWR). As a positive control, we included a high concentration (>1 µg/ml) of two broadly neutralizing human monoclonal antibodies binding distinct epitopes on the SARS-CoV-2 S protein (Clones P2G3 and P5C3), isolated from previously infected and vaccinated donors[68,69]. After incubation, ACE2 mouse Fc fusion protein (produced by the École Polytechnique Fédérale de Lausanne (EPFL) Protein Production and Structure Core Facility) was added to each well at a final concentration of 1 mg/µl and agitated for an additional 60 min. Beads were washed with PBS on a magnetic plate washer (MAG2x program) and 50 µl polyclonal Goat F(ab') anti-mouse IgG-PE secondary antibody (Invitrogen, Catalog 12-4010-87) was added at a 1:100 dilution. Plates were incubated for 45 minutes at room temperature with shaking, washed, washed with PBS and resuspended in 80 µl reading buffer and read on a Bio-Plex 200 plate reader with Bio-Plex Manager software (version 6.2; Bio-Rad). MFI values for beads alone without plasma or antibodies were averaged and used as the 100% binding signal for the ACE2 receptor to the bead-coupled spike trimer. MFI values from wells containing commercial anti-spike blocking antibodies were used as the maximum inhibition signal. Percent blocking of the spike protein trimer:ACE2 interaction was calculated using the formula: %inhibition = 1 − ([MFI test dilution − MFI max inhibition]/[MFI max binding − MFI max inhibition])*100). A lower limit half maximal inhibitory concentration ($IC_{50}$) serum dilution of 50 was set as the

specificity cutoff using $IC_{50}$ values of 104 pre-pandemic healthy donor samples (cutoff$_{50}$ = 12.5 mean $IC_{50}$ + 4*9.0 standard deviation(SD)) to minimize detection of false-positive samples.

### ELISpot Assay

T cell responses were assessed by ELISpot assay using the Human IFN-gamma ELISpot Assay kit (R&D Systems, Catalog EL285) following the manufacturer's instructions. For the assay, cryopreserved PBMCs were thawed in RPMI-1640 (Gibco, Thermo Fisher Scientific) and plated at 5e5 cells per well in in RPMI-1640 medium (Gibco, Thermo Fisher Scientific) supplemented with 5% human AB-serum (BioConcept) and 1% Penicillin-Streptomycin (Thermo Fisher) in assay plates. Cells were stimulated for 20 hours in a humidified incubator at 37 °C and 5% $CO_2$ with overlapping 15mer peptide pools spanning the entire M and N proteins or the S1 domain of the spike protein or a mix of the predicted immunodominant peptides from the spike protein containing the majority of the S2 domain (M, N, S1, and S PepTivator peptide pools, respectively; Miltenyi Biotec). Peptides were dissolved per manufacturer's instructions in sterile water and used at a final concentration of 0.6 nmol (approximately 1 µg/ml) per individual peptide. As unstimulated negative controls, cells were incubated in culture medium alone, without peptide. As positive controls, 2.5e5 cells per well were stimulated with 10 mg/ml anti-CD3 antibody (Clone OKT3, Miltenyi Biotec, Catalog 130-093-387). Longitudinal samples from individual participants were included in the same assay, where possible. Spots were counted using an AID iSpot Reader System with EliSpot 7.0 software (AID). Two times the number of spots in unstimulated negative control wells were subtracted from the values of each test well and results were presented as spot-forming units (SFU) per 1e6 PBMCs with negative values set to zero. Results were excluded if anti-CD3-stimulated positive control wells were negative.

### Flow cytometry and activation-induced marker (AIM) assay

Cryopreserved PBMCs were thawed in RPMI-1640 (Gibco, Thermo Fisher Scientific) supplemented with 5% human AB-serum (BioConcept) and 25U/ml benzonase (Sigma), and plated in 96-UWell plates (Sarstedt) at a concentration of up to 1e6 cells per well in RPMI-1640 medium (Gibco, Thermo Fisher Scientific) supplemented with 10% human AB-serum (BioConcept) and 1% Penicillin-Streptomycin (Thermo Fisher). SARS-CoV-2 PepTivator peptide pools M, N, S1, and S (Miltenyi Biotec) were dissolved per manufacturer's instructions in sterile water and combined into a single megapool. PBMCs were cultured for 24 hours in a humidified incubator at 37 °C and 5% $CO_2$ in the presence of either the SARS-CoV-2 megapool at 0.6 nmol (approximately 1 µg/ml) of each peptide, Phytohemagglutinin-L at 5 µg/ml as a positive control (Merck Millipore) or culture medium (unstimulated condition). Peptide-stimulated and unstimulated samples were run in duplicate whenever possible and longitudinal samples from individual participants were included in the same assay. After 24 hours, cells were washed in staining buffer (PBS, 0.02% $NaN_3$, 2 mM EDTA, 1% bovine serum albumin), blocked for 10 minutes with Human TruStain FcX (Biolegend) on ice and stained for 30 minutes at 4 °C with the following antibodies in buffer supplemented with Super Bright Complete Staining Buffer (eBioscience): BUV395 anti-CD45RA (Clone HI100, BD Bioscience, Catalog 740298, RRID:AB_2740037, Dilution 1:100), BUV496 anti-CD8 (Clone RPA-T8, BD Bioscience, Catalog 612942, RRID:AB_2870223, Dilution 1:100), BUV563 anti-CD56 (Clone NCAM16.2, BD Bioscience, Catalog 612928, RRID:AB_2870213, Dilution 1:50), BUV661 anti-CD14 (Clone M5E2, BD Bioscience, Catalog 741603, RRID:AB_2871011, Dilution 1:100), BUV737 anti-CD16 (Clone 3G8, BD Bioscience, Catalog 564434, RRID:AB_2869578, Dilution 1:100), BUV805 anti-CD19 (Clone SJ25C1, BD Bioscience, Catalog 749173, RRID:AB_2873553, Dilution 1:100), BV421 anti-CD27 (Clone O323, Biolegend, Catalog 302824, RRID:AB_11150782, Dilution 1:50), BV510 anti-CD4 (Clone OKT4, Biolegend, Catalog 317444, RRID:AB_2561866,

Dilution 1:50), BV650 anti-CD38 (Clone HB-7, Biolegend, Catalog 356620, RRID:AB_2566233, Dilution 1:100), BV786 anti-CD3 (Clone OKT3, Biolegend, Catalog 317330, RRID:AB_2563507, Dilution 1:100), PE anti-IgD (Clone IA6-2, Biolegend, Catalog 348204, RRID:AB_10553900, Dilution 1:200), PE/Dazzle594 anti-CCR7 (Clone G043H7, Biolegend, Catalog 353236, RRID:AB_2563641, Dilution 1:50), FITC anti-HLA-DR (Clone L243, Biolegend, Catalog 307604, RRID:AB_314682, Dilution 1:75), PE-Cy7 anti-CD137 (Clone 4B4-1, Biolegend, Catalog 309818, RRID:AB_2207741, Dilution 1:50), BB700 anti-CD134 (Clone ACT35, BD Bioscience, Catalog 746071, RRID:AB_2743451, Dilution 1:50), APC anti-CD69 (Clone FN50, Biolegend, Catalog 310910, RRID:AB_314845, Dilution 1:75). Zombie NIR Fixable Viability Dye (Biolegend) was used to exclude dead cells. After washing, samples were fixed with 1% PFA and acquired on Cytek Aurora 5 L spectral flow cytometer (Cytek). Data was analyzed using Cytek SpectroFlo (version 3.0.1) and FlowJo software (version 10, TreeStar Inc). The gating strategy is demonstrated in Supplementary Fig. 6. Samples were excluded if the percentage of live leukocytes was below 25%. SARS-CoV-2 antigen-specific T cells were measured as AIM$^+$ (CD134$^+$CD137$^+$) CD4$^+$ T and (CD69$^+$CD137$^+$) CD8$^+$ T cells. Unspecific activation in unstimulated controls was subtracted and negative values were set to zero.

## Statistical analyses

We summarized population characteristics descriptively and report frequencies and percentage or median and interquartile range, as applicable. We report summary statistics for the frequency of antibody and T cell responses and calculated 95% Wilson confidence intervals to estimate the associated uncertainty. We excluded any data measured after receipt of COVID-19 vaccination (first dose; $n = 2$ participants at three months and $n = 78$ participants at six months) or diagnosed reinfection (based on self-reported positive PCR or rapid antigen test; $n = 3$ participants at six months) from all analyses. Since we applied an age group-stratified random sampling, we additionally report frequencies for antibody responses after reweighting for the original age group distribution among all SARS-CoV-2-infected individuals in the Canton of Zurich identified during the study period. All data were transformed using natural logarithms for all comparative and associational analyses due to non-normal distributions and ratio properties of the Ig MFI ratios (zero values being replaced by half of the lowest non-zero value). Results are visualized using a log10-transformation in figures.

We validated results of the Luminex assays with corresponding Roche Elecsys assays by calculating the percent concordance and Cohen's Kappa for categorical test results, as well as the Spearman correlation coefficients for absolute measurements. We applied non-parametric Kruskal-Wallis and Friedman tests to examine changes in immune responses over time. While Friedman tests account for the repeated-measures nature of the data, we included both tests since complete follow-up data were not available for all subsample participants. Where reported, we calculated two-tailed p-values without adjustment, with tests considered statistically significant at alpha = 0.05.

To estimate antibody and neutralizing antibody decay times, we first determined the maximum response time point for each individual. We excluded data from individuals that never tested positive for the respective antibody (anti-S IgA, anti-S IgG, anti-N IgG) or in the respective neutralization assay (anti-Wildtype or anti-Delta SARS-CoV-2). We then restricted the data to the maximum and all subsequent timepoints and rescaled the time axis to start with the maximum concentration (MFI ratio or IC$_{50}$), in order to align the data for the descending slope of (neutralizing) antibody decay (in line with previous studies[70,71]). For estimating T cell decay times, we used the data as measured since peak T cell expansion typically occurs in the first week after infection[72]. We then fitted univariable and multivariable mixed-

effects linear decay models, on the natural logarithm-transformed data using random intercepts for individuals. We then calculated the half-life ($\lambda$) in days using the formula, where $\beta$ is the model-derived intercept (and associated uncertainty bounds). We used mixed-model-based parametric bootstrap for the visualization of confidence bounds.

For assessing the correlation of antibody and T cell test results, we calculated Spearman correlation coefficients for all combinations of antibody subtypes and epitope pool-specific T cells, comparing the magnitude of responses as MFI ratios or SFU/1e6 PBMCs, respectively. To assess overall T cell responses, T cell responses specific to M, N, S1, and S2 epitope pools were summed. Furthermore, we assessed concordance by calculating the proportion of participants testing positive or negative for antibody subtypes and overall T cells. Positive antibody responses were defined as an MFI ratio above the limit of detection (see above). Positive T cell responses to individual peptide pools were defined as an SFU value of greater than 0. For overall T cell responses, individuals were considered positive if they were positive to one or more peptide pools. We also assessed agreement by calculating unweighted Cohen's Kappa values for test positivity for all possible combinations of antibody subtypes and virus-specific T cells.

We assessed the association between demographic and clinical factors and antibody (expressed as MFI ratios) and T cell (expressed as SFU / 1e6 PBMCs) responses up to six months after diagnosis using univariable and multivariable mixed-effects linear regression models. Model selection was based on prior knowledge and the Akaike and Bayesian Information Criteria (AIC/BIC), with a difference of 2 points considered relevant. P-values for estimated coefficients were calculated using the t-statistic based on the respective mixed-effects linear regression model derived via Satterthwaite's approximation method for degrees of freedom[50]. Missing data were assumed to be missing completely at random and no imputation was performed. Age, sex, and time since diagnosis were defined as a priori variables based on findings from previous studies. We conducted sensitivity analyses for antibody positivity at two weeks and six months after diagnosis using univariable and multivariable logistic regression analyses, with p-values for estimated coefficients calculated using the z-statistic.

To identify clusters of individuals with similar immune response trajectories over the four assessment timepoints, we used the *KmL3D* k-means clustering method which allows the joint evolution of multiple variables with repeated measures[48]. The algorithm requires predefining the number of clusters. We chose four to six clusters based on a priori knowledge on immune response patterns as well as exploratory analyses of the data. We specified 100 runs for each k clusters (i.e., 300 times in total) and specified Euclidean distance with Gower adjustment to estimate similarity between the trajectories. The selection of the final number of clusters was based on maximizing the Calinski and Harabatz quality criterion[48] as well as the expected patterns in the data. Implementing the *KmL3D* algorithm requires that data for all variables included in the analysis are available for all participants. Hence, missing data were imputed by applying linear extrapolation with added variation (Copy Mean function[48]). We plotted the mean antibody and T cell responses for each cluster to explore differences in respective immune response trajectories. Finally, we descriptively compared demographic and clinical features of individuals in clusters to identify specific factors associated with each trajectory.

All analyzes were performed using R (v4.1.1)[73], using the *Hmisc* (v4.5-0), *psych* (v2.1.6), *survey* (v4.1-1), *lme4* (v1.1-27.1), *lmerTest* (v3.1-3) and *KmL3D* (v2.4.2) packages, and results were visualized using *ggplot2* (v3.3.5), *ggpubr* (v0.4.0), *ggfittext* (v0.9.1), *pheatmap* (v1.0.12), *RColorBrewer* (v1.1-2) and *table1* (v1.4.2) packages.

## Reporting summary

Further information on research design is available in the Nature Research Reporting Summary linked to this article.

## Data availability

All data supporting the findings of this study are available within the paper and its supplementary information files. Source data are provided with this paper.

## Code availability

The analysis code used for the current study (R programming language) is provided in the Supplementary Software file.

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

## Acknowledgements

This study is part of Corona Immunitas research network, coordinated by the Swiss School of Public Health (SSPH+), and funded by fundraising of SSPH+ that includes funds of the Swiss Federal Office of Public Health and private funders (ethical guidelines for funding stated by SSPH+ were respected), funds of the Cantons of Switzerland (Vaud, Zurich, and Basel) and institutional funds of the Universities. Additional funding specific to this study was received from the Department of Health of the Canton of Zurich, the University of Zurich Foundation, and the Swiss Federal Office of Public Health. The funding bodies had no influence on the study design or the conduct, analysis, or interpretation of the study findings. T.B. received funding from the European Union's Horizon 2020 research and innovation programme under the Marie Skłodowska-Curie grant agreement No 801076, through the SSPH+ Global PhD Fellowship Programme in Public Health Sciences (GlobalP3HS) of the SSPH+. H.E.A. received a Swiss National Science Foundation (SNSF) Early Postdoc.Mobility Fellowship. Furthermore, the authors would like to thank Danusia Vanoaica and Osman Yoztekin for their support with the T cell assays, and the study administration team and the study participants for their dedicated contribution to this research project.

## Author contributions

Conception and design: D.M., K.D.Z., T.B., C.M., M.A.P., and J.S.F. Development of methodology: D.M., K.D.Z., T.B., N.C., C.P., M.P., C.F., G.P., C.M., M.A.P., and J.S.F. Funding acquisition: M.A.P. and J.S.F. Supervision: C.M., M.A.P., and J.S.F. Project administration: D.M., K.D.Z., T.B., H.E.A., A.D., M.A.P., and J.S.F. Data acquisition: D.M., K.D.Z., T.B., N.C., D.L.C., C.P., and M.P. Analysis and interpretation of data: D.M.,

K.D.Z., T.B., N.C., C.P., M.P., C.F., G.P., C.R.K., C.M., M.A.P., and J.S.F. Data visualization: D.M., K.D.Z., T.B., and N.C. Writing of original manuscript draft: D.M., K.D.Z., and T.B. Review and editing of manuscript: D.M., K.D.Z., T.B., N.C., D.L.C., H.E.A., A.D., C.P., M.P., C.F., G.P., C.R.K., C.M., M.A.P., and J.S.F. All contributing authors approved the submitted manuscript.

## Competing interests

The authors declare no competing interests.
