## [Peer Review File · Nature Communications]

Heterogenous Humoral and Cellular Immune Responses with Distinct Trajectories Post-SARS-CoV-2 Infection in a Population-Based CohortREVIEWER COMMENTS

Reviewer #1 (Remarks to the Author):

This study by Menges and colleagues conducted a longitudinal human cohort study on over 400 individuals to assess the development of antibody responses to the S and protein as well as for a subset the anti-NP IgG response and T cell responses to three SARS-CoV2 proteins over a roughly 6 months period following infection. The study overall describes distinct profiles of T and antibody responses that are overall tracking with clinical features and contribute to and support previous conclusions that older adults with more severe disease generated more robust IgG and IgA as well as T cell responses, compared to those with milder disease, and demonstrate overall a modest decay for both antibody and T cell responses over the study period.

Overall, the study design, methods and data analysis used is of very high quality and the manuscript is clearly written and figures are sufficient for the reader to interpret the data shown. Supplemental data are important for data interpretation. Although the study is not breaking new ground, it provides important clinical data on a cohort of infected patients, including an analysis of CD4 and CD8 T cell responses. An inherent limitation of the study is of course the analysis of blood only. Given the otherwise careful discussion of study limitations, it was surprising that the authors did not consider redistribution of T cell populations to contribute to the somewhat surprising increase in responses in one of the clusters. However, based on the discussion these patients seem to have had secondary clinical symptoms not captured perhaps in the tables –discussion on the likelihood or not of immune cell redistribution to explain result fluctuations should be considered.

Additional comments

1. Lines 106 – 110 seem superfluous given the supplemental tables and Fig. 1
2. Given that the subsample was deliberately chosen to reflect the full spectrum of patients, the word “random” should probably not be used (Figure 1B) to describe the subset of 64 patients.
3. Fig. 4B would be much clearer if the lines could be color-coded
4. Fig. 5B needs a color-legend

Reviewer #2 (Remarks to the Author):

Dear authors,

in this study, the authors performed longitudinal analyses of SARS-CoV-2 specific IgA and IgG antibodies of 431 infected individuals up to 6 months after diagnosis of infection. In a subgroup of 64 persons, they additionally analyzed nucleoprotein-specific IgG as well as T-cell immune responses towards peptides of the SARS-CoV-2-spike, membrane and nucleocapsid protein. Based on these analyses, the authors identified 5 immune trajectories differing in the progression of the specific immune responses.

The manuscript is well written and illustrates the data in a clear and comprehensible way. Although there are several other studies in the literature dealing with SARS-CoV-2 specific immunity after infection, the authors beyond that present a very detailed and interesting view on how individuals' immune systems differentially react to SARS-CoV-2 infection and make some connections with potential causes of these differences by combining the data with demographic and clinical factors. Nevertheless, I have the following comments that should be addressed to increase clarity.

Major comments:

- Information on critically ill patients is very limited (only two patients in the cohort requiring ICU admission). This should be emphasized more clearly in the discussion.
- Please provide information on the numbers of analyzed data sets at each time-point more clearly in the main manuscript (especially in the figure legends) and not only in the supplementary tables (see

numbers in Table S2 and S3).

- p.8 l.166: If there are not too many missing data sets, the respective paired analysis (Friedman test) including the individuals with samples at all 4 time points may be more representative than Kruskal-Wallis testing. (This may have consequences for the next comment, too.)
- p.8 l.170-172: The stable level of activated CD8 T cells in the AIM+ assay seems overemphasized in comparison to the slight but not significant decrease in activated CD4 T cells. Consider rephrasing.
- p.22 l.514-516 and Fig. 3A): There is a discrepancy in the description of the methods (“...results were normalized to the number of spots in anti-CD3 antibody-stimulated wells... and presented as spot-forming units (SFU) per 1e6 CD3+ cells”) and the presentation in Fig. 3A (as “Spec. TC/1e6 PBMCs). Please clarify.
- Fig. 3: Can the authors provide information on paired analyses of the results depicted in Panel A and D (see also comment 3). This may give a better impression on the individual courses of T-cell responses.
- Fig. 3A: Please show the medians (+IQR?).
- Fig. 3D/E: Medians (+IQR?) may be more representative of the data than the indicated means, especially as the authors use non-parametric testing for analysis of the data. Please consider revising accordingly.

Minor comments:

- p.6 l.123-126: As the whole study population and the subsample slightly differ in antibody distribution, it would be interesting to know the calculated half-life of anti-S-IgA and -IgG in the subsample to get an impression on comparability of the already mentioned half-life values (anti-S-IgA and -IgG only in whole study population, anti-N-IgG in the subsample).
- p.8 l.184 and Fig. 4A: I suggest the expression “pooled M, N, S1, S2” refers to the “Any TC” in the figure (which means T cells reacting to any of the different stimuli in contrast to T cells reacting to the pooled stimuli in one stimulation reaction). Please rephrase to avoid confusion (especially when compared to the AIM+ assay where the peptides of the different virus proteins were indeed used for a combined stimulation).
- Legend to Fig. 2C: Typo: “anti-S-IgG” has to be “anti-N-IgG”
- Fig. 3E: Switching the CD4 and CD8 panels (to be congruent to Fig. 3D) would be more intuitive.
- Fig. 5B: Please provide information on color coding (only the grey color is explained so far)
- Please provide a figure depicting the gating strategy of the flow-cytometrical analysis (in the supplement).

Reviewer #3 (Remarks to the Author):

This is an interesting study analyzing individual long-term immune responses following SARS-CoV-2 infection in a representative randomly selected adult cohort with mostly mild disease. The study design is robust, with clearly defined sampling time points. In general, the study would benefit from a broader spectrum of immunological readouts and a larger subcohort undergoing more detailed immunological assays, especially when they are divided into 5 clusters.

- The most important limitation is the utilization of a single SARS-CoV-2 serology assay, without other readouts of the humoral immune response
- In the methods section, the authors describe a commercial pooled serum sample as a negative control. It would be important to show the negative control values, ideally not from one pooled, but from several individual pre-pandemic control samples in Figure 2.
- The authors use a cross-validation to convert the MFI ratios into the standard unit BAU/ml. How was this formula generated?
- Almost 20% of all PCR-proven infected participants of the study cohort are seronegative for antiS-IgG (Table S2), despite serial testing. This is surprisingly low.
- In the subsample of 64 individuals which underwent T cell assays, the seropositivity is even lower at around 60%. The authors should comment on this – this is a lot lower than expected. Downstream data from a cohort where 4 out of 10 participants did not seroconvert for SARS-CoV-2 in several

serial assays despite PCR-proven infection are hard to interpret. The referenced paper (45) for the method shows a sensitivity of 97% >d16 post symptom-onset (Fenwick, J Virol 2021, Fig 1)?

- I would recommend to use at least one orthogonal commercial SARS-CoV-2 serology assay to better understand the sensitivity and specificity of the Luminex assay

- It would be interesting to get more insights into the humoral immune response, looking at variants of concern and equally important neutralization capacity rather than antibody levels to the Wuhan strain alone (that is no longer circulating)

Figure 5:

- The approach to identify clusters in this subgroup is clearly interesting. However, given the sample size (n=64), subsets of 8-9 participants are small, so one should be careful with drawing firm conclusions. This should be brought to the readers' attention more directly (rather than talking about relative values of the subsets in the text) in the manuscript and in the figure legends.

- Overall, the panels 5A-5C are somewhat redundant as they are showing the same data in different ways. This should be condensed. The Ab decay in the S-IgG ratio looks a lot more pronounced in Fig 5A, whereas the differences in 5C are very subtle

- Given the small subgroups I would also recommend to discuss these data in Fig. 5 more compact in the manuscript

- Figure 6: There seems to be an age effect with older participants having higher antibodies. However, the age effect could be rather a severity effect, with older people having more severe disease. This shows that the strong interdependence of the variables. These effects should be analyzed (I speculate severity of disease is the main driver), the current visualization does not account for these biases and can be misunderstood

- L409 "None, however, tested positive. This again highlights the role of the immune system in protecting from severe disease upon re-exposure,..." This is speculation. If the SARS-CoV-2 test (PCR?) was negative, how does this then highlight the role of the immune system for protection of disease? Apparently the reason for the test is unknown (contact tracing vs. symptomatic episode), so this conclusion is not possible. Also, this fact that the participants got retested with a negative result does not demonstrate that "reinfections are more frequent than reported".

Point-by-Point Response to Reviewers' Comments

Reviewer #1:

This study by Menges and colleagues conducted a longitudinal human cohort study on over 400 individuals to assess the development of antibody responses to the S and protein as well as for a subset the anti-NP IgG response and T cell responses to three SARS-CoV2 proteins over a roughly 6 months period following infection. The study overall describes distinct profiles of T and antibody responses that are overall tracking with clinical features and contribute to and support previous conclusions that older adults with more severe disease generated more robust IgG and IgA as well as T cell responses, compared to those with milder disease, and demonstrate overall a modest decay for both antibody and T cell responses over the study period.

Overall, the study design, methods and data analysis used is of very high quality and the manuscript is clearly written and figures are sufficient for the reader to interpret the data shown. Supplemental data are important for data interpretation. Although the study is not breaking new ground, it provides important clinical data on a cohort of infected patients, including an analysis of CD4 and CD8 T cell responses. An inherent limitation of the study is of course the analysis of blood only. Given the otherwise careful discussion of study limitations, it was surprising that the authors did not consider redistribution of T cell populations to contribute to the somewhat surprising increase in responses in one of the clusters. However, based on the discussion these patients seem to have had secondary clinical symptoms not captured perhaps in the tables –discussion on the likelihood or not of immune cell redistribution to explain result fluctuations should be considered.

Thank you very much for your valuable comments. We agree that the use of blood only is certainly a limitation, particularly as it is increasingly well-understood that a substantial portion of the “action” in terms of the adaptive immune response (particularly for T cell responses, but of course also for antibody responses and newly described tissue-resident B cells) occurs at sites of infection. We would like to mention that we were unable to collect other samples (such as nasal washes) due to SARS-CoV-2 being classified as a BSL-3 organism at the time of study initiation. The issue of redistribution from the circulation to the tissues (and vice versa) is an excellent point and we have alluded to this briefly in our discussion (but have also tried not to overstep our “boundaries” as we only were able to analyze blood samples for this study and can only speculate). The relevant text is as follows:

p.18 line 430 “Of note, about half of cluster participants underwent SARS-CoV-2 testing between three and six months, suggestive of exposure events or symptomatic episodes. None, however, reported testing positive. It is also possible that, rather than re-expansion, the decrease at one to three months reflects trafficking of lymphocytes from the circulation into the tissues. Indeed, levels of total CD4⁺ T cells, B cells and NK cells also tended to be lower in this cluster at these timepoints (Supplementary Fig. 4b). Studies of lung lavage and tissue samples from COVID-19 patients have demonstrated expression of tissue homing and tissue residence marker expression on T cells (Grau-Expósito et al., Nat Commun 2021; Saris et al., Thorax 2021; Liao et al., Nat Med 2021; Poon et al., Sci Immun 2021). As we were only able to obtain blood from participants, we were unable to address the issue of site-specific immunity in the current study, though a better understanding of the spatial nature of the immune response to infection is certainly necessary.”

Additional comments

1. Lines 106 – 110 seem superfluous given the supplemental tables and Fig. 1

We agree and have removed this paragraph.

2. Given that the subsample was deliberately chosen to reflect the full spectrum of patients, the word “random” should probably not be used (Figure 1B) to describe the subset of 64 patients.

This is a good point and we agree that this may be confusing for the reader. To clarify, we performed random sampling within several predefined strata (i.e., low and high disease severity, low and high antibody responses, hospitalization, age group, sex) to select participants for this sample – but it is not a “random sample” in the strict sense. To prevent this from being misleading we have modified the wording from “Random subsample covering full spectrum of infected population” to “Selected subsample” and “... subsample of participants selected to cover the spectrum of infection” in the figure and caption, respectively. We have made sure to use this terminology throughout the paper. We also include details regarding the selection of participants for the subsample in the Methods section.

3. Fig. 4B would be much clearer if the lines could be color-coded

We have implemented this suggestion. Please see the revised Fig. 4b, which we agree helps make the figure much easier to interpret.

4. Fig. 5B needs a color-legend

Thank you for bringing this to our attention. We have now added the color legend to Fig. 5b.

Reviewer #2:

Dear authors,

in this study, the authors performed longitudinal analyses of SARS-CoV-2 specific IgA and IgG antibodies of 431 infected individuals up to 6 months after diagnosis of infection. In a subgroup of 64 persons, they additionally analyzed nucleoprotein-specific IgG as well as T-cell immune responses towards peptides of the SARS-CoV-2-spike, membrane and nucleocapsid protein. Based on these analyses, the authors identified 5 immune trajectories differing in the progression of the specific immune responses.

The manuscript is well written and illustrates the data in a clear and comprehensible way. Although there are several other studies in the literature dealing with SARS-CoV-2 specific immunity after infection, the authors beyond that present a very detailed and interesting view on how individuals' immune systems differentially react to SARS-CoV-2 infection and make some connections with potential causes of these differences by combining the data with demographic and clinical factors. Nevertheless, I have the following comments that should be addressed to increase clarity.

Major comments:

- Information on critically ill patients is very limited (only two patients in the cohort requiring ICU admission).

This should be emphasized more clearly in the discussion.

First, we would like to thank you for your helpful comments and for taking the time to review our paper.

Regarding the above point, we agree that this should be better described. We feel that one of the strengths of our study is that it looks at the immune response in a population-based manner, possibly indicating desirable immune responses for virus clearance without severe disease. As compared to hospital-based studies, though, we may “miss” the most critically-ill patients. Without a doubt, critically-ill individuals likely show distinct immune response patterns from the broader population of infected individuals. To address this relevant point, we have included the following text in the discussion. We also have modified our description “full spectrum of the disease” to “spectrum of the disease” throughout the text to be consistent with this.

“First, we captured relatively few critically-ill patients in our study, which may show distinct immune response patterns (Lucas et al., Nature 2021; Giamarellos-Bourboulis et al., Cell Host Microbe 2020; Mathew et al., Science 2020).”

- Please provide information on the numbers of analyzed data sets at each time-point more clearly in the main manuscript (especially in the figure legends) and not only in the supplementary tables (see numbers in Table S2 and S3).

We have now added the number of measurements at each timepoint to the figure legends in Fig. 2a-f and Fig. 3a-d in line with your suggestion. We hope this will now make the number of samples analyzed for each assay at each timepoint more transparent for readers.

- p.8 l.166: If there are not too many missing data sets, the respective paired analysis (Friedman test) including the individuals with samples at all 4 time points may be more representative than Kruskal-Wallis testing. (This may have consequences for the next comment, too.)

Thank you for bringing this up, since this was a point that we also were unsure about in our initial analyses. We agree that Friedman tests are a better choice if there are not too many missing data. For our overall and antigen-specific T cell analyses, we had relatively complete data (samples/enough cells from all four timepoints) for 75% of participants. For the CD4⁺ and CD8⁺ T cell AIM assays, we had complete data for 55% as, in some instances, we no longer had sufficient PBMCs for some participants at certain timepoints. Due to these missing data, we felt the Friedman test may not be the optimal choice as our sample size was considerably reduced. We then elected to use Kruskal-Wallis testing to keep the type of analysis consistent throughout. However, the results are comparable with both tests. We now provide the results of both tests for transparency. In addition, we have made the number of measurements at each timepoint clearer within the figure legends as discussed above.

- p.8 l170-172: The stable level of activated CD8 T cells in the AIM+ assay seems overemphasized in comparison to the slight but not significant decrease in activated CD4 T cells. Consider rephrasing.

It is true that the decline is not especially prominent and it is certainly likely that even if the AIM⁺CD4⁺ T cell population did decline over time that it could still be recalled upon subsequent viral exposure and play an important function for later protection. We do still wish to point the decline out, but we would rather highlight that the two T cell populations may serve different roles. For example, that AIM⁺CD8⁺ T cells have a more dominant TEMRA phenotype compared to the more dominant TCM phenotype of AIM⁺CD4⁺ T cells. We have adjusted this in the results and discussion as follows:

Results – “For CD4⁺ T cells, this was 0.1% at six months, though this decline did not reach statistical significance ($p=0.11$, Kruskal-Wallis test; $p=0.14$, Friedman test; Fig. 3d). In contrast, for CD8⁺ T cells, frequencies of AIM⁺ cells remained at 0.2% up to six months post-diagnosis. In assessing the

phenotype of AIM⁺ T cells in the blood, AIM⁺CD4⁺ T cells were predominantly central memory T cells (TCM), while AIM⁺CD8⁺ T cells were predominantly T effector memory cells re-expressing CD45RA (TEMRA; Fig. 3e). Taken together, our results suggest that anti-S-IgG, along with N- and S2-specific T cells, may act as more long-lasting components of SARS-CoV-2-specific immunity following infection, and that circulating virus-specific CD4⁺ and CD8⁺ T cells tend to provide protection through more TCM- and TEMRA-like functions, respectively.”

Discussion – “In assessing virus-specific CD4⁺ and CD8⁺ T cells by AIM assay, we found that frequencies of AIM⁺CD4⁺ or AIM⁺CD8⁺ T cells were similar at two weeks post-diagnosis, but that activated, virus-specific CD4⁺ T cells appeared to decline slightly over the six months of follow-up compared to AIM⁺CD8⁺ T cells. Furthermore, AIM⁺CD4⁺ T cells were predominantly of TCM phenotype, and this was consistent throughout the convalescent period, whereas AIM⁺CD8⁺ T cells had a predominantly TEMRA phenotype, in agreement with previous studies. These findings support the idea that virus-specific CD4⁺ T cells may play an important role in maintaining the immune response and immunological memory, whereas CD8⁺ T cells may play a more direct role in the anti-viral immune response as highly activated and more terminally-differentiated cytotoxic lymphocytes.”

- p.22 | 514-516 and Fig. 3A): There is a discrepancy in the description of the methods (“...results were normalized to the number of spots in anti-CD3 antibody-stimulated wells... and presented as spot-forming units (SFU) per 1e6 CD3⁺ cells”) and the presentation in Fig. 3A (as “Spec. TC/1e6 PBMCs). Please clarify.

Thank you for pointing out this oversight. In our initial analyses, we normalized counts to the number of CD3⁺ cells. However, to be more consistent with other published literature, we ultimately chose to present our data as SFU/1e6 PBMCs. We have fixed this in the methods (which now reads as “Two times the number of spots in unstimulated negative control wells were subtracted from the values of each test well and results were presented as spot-forming units (SFU) per 1e6 PBMCs with negative values set to zero.”) and have also adjusted this in Figure 3.

- Fig. 3: Can the authors provide information on paired analyses of the results depicted in Panel A and D (see also comment 3). This may give a better impression on the individual courses of T-cell responses.

As mentioned, we had previously selected Kruskal-Wallis testing based on missing values (particularly in the AIM analysis datasets). However, we now also provide Friedman test results (please see the revised Fig. 3 a/d). We would like to mention that, due to the heterogeneity within T cell responses between individuals, we feel the cluster analyses are probably of special interest here.

-Fig. 3A: Please show the medians (+IQR?).

We agree that the medians and interquartile ranges provide a better representation of the data than the means and adapted Fig. 3a/d, Fig. 5c and Supplementary Fig. S4c to include boxplots indicating median and IQR (see also next comment). For Fig. 3a, this required changing the x-axis from (continuous) time since diagnosis to (categorical) timepoints. The data previously shown in Fig. 3a are still presented within Supplementary Fig. S2d-h as part of the T cell decay analyses.

- Fig. 3D/E: Medians (+IQR?) may be more representative of the data than the indicated means, especially as the authors use non-parametric testing for analysis of the data. Please consider revising accordingly.

As mentioned, we have adapted Fig. 3a/d, Fig. 5c and Supplementary Fig. S4a to present medians and interquartile range. We did indeed mostly rely on non-parametric testing in our analyses. Meanwhile, we would like to add that we also used linear regression modeling of natural logarithm-transformed data, for which the distributions should be adequate to apply parametric testing.

Minor comments:

- p.6 | 123-126: As the whole study population and the subsample slightly differ in antibody distribution, it would be interesting to know the calculated half-life of anti-S-IgA and -IgG in the subsample to get an impression on comparability of the already mentioned half-life values (anti-S-IgA and -IgG only in whole study population, anti-N-IgG in the subsample).

This is an important comment since there are indeed some relevant differences between the overall study population and the subsample, for which we selected participants both with high and low antibody responses. In line with your suggestion, we calculated the half-life for anti-S-IgA and -IgG within the subsample using the same approach as with the full population. We find similar results in these sensitivity analyses (though 95% confidence intervals were larger, as expected). To make this more transparent we have added Supplementary Table S5, which provides the detailed results for the estimated half-lives for antibody responses, neutralization capacity, and T cell responses based on unadjusted and adjusted decay modeling.

- p.8 | 184 and Fig. 4A: I suggest the expression “pooled M, N, S1, S2” refers to the “Any TC” in the figure (which means T cells reacting to any of the different stimuli in contrast to T cells reacting to the pooled stimuli in one stimulation reaction). Please rephrase to avoid confusion (especially when compared to the AIM+ assay where the peptides of the different virus proteins were indeed used for a combined stimulation).

Yes, you are correct in this interpretation and we agree that this was confusing as written. We have rephrased this in the figures, as well as throughout the results, figure captions, and in the methods to make this distinction clearer. We further made sure that we use the same terminology consistently throughout the manuscript.

- Legend to Fig. 2C: Typo: “anti-S-IgG” has to be “anti-N-IgG”

Thank you for pointing out this mistake. We have changed the legend for Fig. 2c to "anti-N-IgG".

Fig. 3E: Switching the CD4 and CD8 panels (to be congruent to Fig. 3D) would be more intuitive.

We switched the panels according to your suggestion.

- Fig. 5B: Please provide information on color coding (only the grey color is explained so far)

Thank you for bringing this to our attention. We have added a color legend to Fig. 5b.

- Please provide a figure depicting the gating strategy of the flow-cytometrical analysis (in the supplement).

We have added a depiction of the FACS gating strategy in the supplement (see Supplementary Fig. S5).

Reviewer #3:

This is an interesting study analyzing individual long-term immune responses following SARS-CoV-2 infection in a representative randomly selected adult cohort with mostly mild disease. The study design is robust, with clearly defined sampling time points. In general, the study would benefit from a broader spectrum of immunological readouts and a larger subcohort undergoing more detailed immunological assays, especially when they are divided into 5 clusters.

- The most important limitation is the utilization of a single SARS-CoV-2 serology assay, without other readouts of the humoral immune response

Thank you very much for your insightful comments. We agree with you and also felt that it was a valuable addition to use another method to validate our findings. Therefore, we additionally tested all samples/timepoints from subsample participants using two commercially available assays which detect anti-SARS-CoV-2 S- or N-specific total Ig (Roche Elecsys® Anti-SARS-CoV-2 S and Roche Elecsys® Anti-SARS-CoV-2, run on the Roche Cobas e411 instrument), respectively. We observed a high percent agreement (overall 98.3% for anti-S-Ig or anti-N-Ig seropositivity; now included in the revised manuscript as Supplementary Fig. S1c–d) and a high degree of correlation with the corresponding Luminex-based tests (Spearman $r=0.89$ for anti-S-Ig, Spearman $r=0.86$ for anti-N-Ig; now included as Supplementary Fig. S1e in the revised manuscript) which help to confirm our findings. The additional testing thus confirms our previous findings. In addition, we now provide additional external cross-validation data for the Luminex and Roche Elecsys® Anti-SARS-CoV-2 S tests based on data from infected individuals that were not part of this study (see comment below), which further demonstrates the high correlation between tests. One caveat that we would like to mention is that the Roche tests do not distinguish between Ig subtypes and is targeted at the RBD domain of the Spike protein only (whereas the Luminex test is targeted at the trimeric Spike protein). This could contribute to the residual differences observed between Roche and Luminex tests and is one of the reasons that we preferred to use the Luminex assay for the main analysis.

- In the methods section, the authors describe a commercial pooled serum sample as a negative control. It would be important to show the negative control values, ideally not from one pooled, but from several individual pre-pandemic control samples in Figure 2.

Here we would like to clarify that the MFI ratios which we present are generated by dividing the raw luminescence of tested samples by the raw luminescence of the control pooled serum sample included on each assay plate to normalize the measurements. This standardization is somewhat analogous to the way which we subtract background (unstimulated) values from stimulated test samples assessed in ELISpot or AIM assays (which are also not shown as non-normalized values as this would be confusing and not really meaningful). As the control serum is pre-pandemic, we make the necessary assumption that it does not contain any SARS-CoV-2-specific antibodies and that whatever raw luminescence value that is generated corresponds to the “negative” baseline (unfortunately individuals cannot serve as their own internal controls here as we do not have blood samples from before they contracted SARS-CoV-2, which would be ideal).

We realized, though, based on your comment, that we did not clearly indicate the limit of detection of the assay in our figures such that the reader could understand the cutoff for what qualifies as a “positive” test result (we included as dashed lines on each plot, though we failed to indicate/discuss this in the first version of the manuscript). We have now added this information and hope that this aids in the interpretation of our results. Just in case that you are interested, the limit of detection cutoff values were determined in a previous study validating the Luminex assay with pre-pandemic control

samples (Fenwick et al., J Virol 2021). Therein, cutoffs for mean fluorescence ratio values were defined at 6.5 for IgA and 6.0 for IgG, which we continue to use in our study.

- The authors use a cross-validation to convert the MFI ratios into the standard unit BAU/ml. How was this formula generated?

In addition to the initial validations (Fenwick et al., J Virol 2021), additional cross-validations were performed on 900 samples from SARS-CoV-2 infected individuals of the Lausanne University Hospital (Switzerland; participants were not part of the reported population-based study), which were tested with both the Luminex anti-S-IgG assay and the Roche Elecsys® anti-SARS-CoV-2 S assay. We now report data from these additional external validations in Supplementary Fig. S1f. Within these data, we used a linear regression model of $\log_{10}(\text{U/ml} - \text{Elecsys}^{\circledR})$ values as a function of $\log_{10}(\text{MFI ratio} - \text{Luminex})$ values. We then used this linear function to estimate conversion values for Luminex MFI ratios (corresponding to the red line in Supplementary Fig. S1f). Of course, this conversion formula can only provide an approximation. However, we felt that the conversion provides a valuable addition to the manuscript, since it allows a better comparison of our results in the overall study population with other studies.

- Almost 20% of all PCR-proven infected participants of the study cohort are seronegative for anti-S-IgG (Table S2), despite serial testing. This is surprisingly low.

We also thought that this was an interesting finding. In our total study sample, 14.8% of participants were never seropositive for anti-S-IgG (15.5% for anti-S-IgA, 13.7% for either up to 3 months). Studies that have found higher levels of seropositivity typically include primarily hospitalized or severely affected individuals. As our study utilizes a population-based sample of SARS-CoV-2 infected individuals we include higher proportions of mild disease cases and asymptomatic infections caught through contact testing. Since higher disease severity is associated with stronger immune responses (which we see here as well), it is perhaps not surprising that we observe lower levels of seropositivity. While few well-designed population-based studies exist, these have also reported proportions of seropositivity around 83-91%, similar to what we see here (e.g. Vanshylla et al., Cell Host & Microbe 2021; Gudbjartsson et al., NEJM 2020).

In addition, as discussed above, in using a second set of assays to confirm our findings, we observed an overall agreement of 98.3% for anti-S-Ig or anti-N-Ig within the subsample, lending support to our findings. We cannot rule out, of course, that individuals have only very low levels or different subtypes/specificities of circulating antibodies or that they have mucosal but not circulating antibody responses, which we would not be able to detect in our study.

- In the subsample of 64 individuals which underwent T cell assays, the seropositivity is even lower at around 60%. The authors should comment on this – this is a lot lower than expected. Downstream data from a cohort where 4 out of 10 participants did not seroconvert for SARS-CoV-2 in several serial assays despite PCR-proven infection are hard to interpret. The referenced paper (45) for the method shows a sensitivity of 97% >d16 post symptom-onset (Fenwick, J Virol 2021, Fig 1)?

This is a very important point. For the subsample, we aimed to cover the full spectrum of disease severity and immune responses when selecting the participants. The reason for this was that we were interested in assessing whether there would be differences in T cell response patterns between seronegative and seropositive individuals (and between severe and less severe disease). We selected individuals at random within strata of disease severity and levels of antibody responses (along with hospitalization, age group, sex). Through this process, participants with very low antibody

responses were oversampled. This inevitably resulted in a lower overall seropositivity (66% ever testing anti-S-IgG positive). This was within the range that we expected, though, given that overall seropositivity was approximately 85%. We added a statement to the revised version of the manuscript to help make this clearer (lines 138-140: “These lower proportions of seropositivity in the subsample were expected, as we aimed to specifically include individuals with low antibody responses in our assessment of T cell responses.”). We also have modified the wording in Fig. 1b and its caption from “Random subsample covering full spectrum of infected population” to “Selected subsample” and “... subsample of participants selected to cover the spectrum ...”, respectively. We have made sure to use this terminology throughout the paper and provide details regarding the selection of participants for the subsample in the Methods section.

To address the serological testing; indeed, the Luminex assay used in this study showed a sensitivity of 97% in validation studies and performed better than five common commercial assays (range 83-93%; Fenwick et al., J Virol 2021, Fig. 3). Importantly, in the validation, most of the PCR-/sero-positive individuals that were included were hospitalized or severely affected. In our sample, all hospitalized participants and 93% of severely affected participants were also anti-S-IgG positive. Additionally, we found an overall agreement of over 98% for anti-S-Ig or anti-N-Ig positivity between the Luminex and Roche assays, lending support to the idea that these individuals do not develop detectable circulating antibody responses, which is in line with other work in mildly affected individuals (see also Vanshylla et al., Cell Host & Microbe 2021). However, as we have mentioned, it is entirely possible that these individuals still develop very low circulating responses or mucosal antibody responses which could be detected in, for example, nasal washes. Unfortunately, we were not able to collect/assess such samples for this study due to biosafety level constraints as SARS-CoV-2 was classified as a BSL-3 organism in Switzerland at the time of study initiation.

- I would recommend to use at least one orthogonal commercial SARS-CoV-2 serology assay to better understand the sensitivity and specificity of the Luminex assay

We agree and have implemented your suggestion as described above and in the revised version of the manuscript. Please see Supplementary Fig. S1c–f.

- It would be interesting to get more insights into the humoral immune response, looking at variants of concern and equally important neutralization capacity rather than antibody levels to the Wuhan strain alone (that is no longer circulating)

We fully agree that neutralizing capacity towards the emerging variants of concern has become highly relevant in recent months. So far, only few studies have investigated neutralizing capacity repeatedly over time (e.g. Wang et al., CID 2021; Wajnberg et al., Science 2020; Feng et al., Nature Comm 2021). In the revised version of the manuscript, we include neutralization activity data for all individuals in the subsample of 64 individuals over the 6 months of observation. We used a novel high-throughput assay (Fenwick et al., Science Transl Med, 2021; <https://doi.org/10.1126/scitranslmed.abi8452>) to test for neutralizing antibodies against Wildtype-SARS-CoV-2 as well as the Delta and Omicron variant (see Methods lines 556-579 for details). We now include a descriptive analysis of neutralization capacity and an estimation of the neutralizing antibody decay in the manuscript (see new Fig. 2d-f and Fig. 2j-k). Overall, about 45% (70% of those who were seropositive) developed anti-Wildtype neutralizing antibodies and 15% (23% of seropositives) anti-Delta neutralizing antibodies. Meanwhile, neutralizing activity against Omicron was observed in only one of the participants (just to note – decay analysis was not possible for Omicron for this reason), in line with what would be expected among infected

individuals without re-exposures through vaccination or reinfection. In decay analyses, we found that half-lives of neutralizing antibodies against Wildtype-SARS-CoV-2 and the Delta variant are shorter than for total antibody responses. In the identified clusters, trends paralleled those of the overall antibody trajectories (i.e., anti-S-IgG and anti-N-IgG). We strongly feel that this additional data substantially increases the value of our work and hope that you agree.

Figure 5:

- The approach to identify clusters in this subgroup is clearly interesting. However, given the sample size (n=64), subsets of 8-9 participants are small, so one should be careful with drawing firm conclusions. This should be brought to the readers' attention more directly (rather than talking about relative values of the subsets in the text) in the manuscript and in the figure legends.

Definitely, thank you – we have now added the n for each cluster in the main text as well as in the legend for Fig. 5 (added: "(total n=64; cluster 1: n=9, cluster 2: n=8, cluster 3: n=12, cluster 4: n=10, cluster 5: n=25)"). We have also added the following text in the discussion (lines 410-415): "... We observed that clusters (each containing between eight and 25 individuals) independently demonstrated ...".

- Overall, the panels 5A-5C are somewhat redundant as they are showing the same data in different ways. This should be condensed. The Ab decay in the S-IgG ratio looks a lot more pronounced in Fig 5A, whereas the differences in 5C are very subtle

Thank you for pointing this out. We have now replaced the redundant plots (anti-S-IgG, N-TC, S2-TC) in Fig. 5c with new plots depicting the trajectories of neutralizing capacity (anti-Wildtype and anti-Delta IC₅₀) as well as AIM⁺CD4⁺ and AIM⁺CD8⁺ T cell frequencies. We now include the information previously shown in Fig. 5c in Supplementary Fig. S4a, where we have also added a plot for anti-Omicron neutralizing capacity.

Regarding the appearance of the antibody decay; to perform the clustering analysis the immune response data had to be normalized (rescaled). These normalized values are shown Fig. 5a. We feel this representation provides a good depiction of the joint immune trajectories in the different clusters as antibody and T cell responses are presented on the same scale. For transparency, however, we wanted to include the measured results by cluster as well (as shown in Fig. 5b and Supplementary Fig. S4a).

- Given the small subgroups I would also recommend to discuss these data in Fig. 5 more compact in the manuscript

We agree that this section was somewhat long and have shortened the text regarding immune response trajectories within clusters. While we still discuss each cluster individually to point out potentially interesting features, we aimed to do so in a more condensed way. Please see the text in lines 249-317.

- Figure 6: There seems to be an age effect with older participants having higher antibodies. However, the age effect could be rather a severity effect, with older people having more severe disease. This shows that the strong interdependence of the variables. These effects should be analyzed (I speculate severity of disease is the main driver), the current visualization does not account for these biases and can be misunderstood

This is an important point which we are happy to clarify. Indeed, it is likely that there is some interdependence between higher disease severity and age. However, we have already considered and accounted for this in our analyses. The results in Fig. 6a-d are based on multivariable regression models (by definition every variable in the model is mutually adjusted for, so that one can identify “independent” associations). The models which we have used are adjusted for time since diagnosis, age group, sex, and disease severity. Therefore, the results that we present for age and disease severity are mutually accounted for and can be interpreted as, for example, “effects for age, independent of disease severity” (and vice versa).

For interested readers, we have also added the results from univariable (unadjusted) analyses to the Supplementary Material (Tables S6-S8). These show the associations without accounting for other factors. In this way one can now see the changes in estimates before and after mutual adjustment. Interestingly, associations for age and disease severity with antibody and T cell responses become even stronger after adjustment. Consistent with our findings here, other groups have also found age and disease severity to be independently associated with the strength of immune responses (e.g. Gudbjartsson et al., NEJM 2020; Hansen et al., J Immun 2020; Vanshilla et al., Cell Host & Microbe 2021).

- L409 “None, however, tested positive. This again highlights the role of the immune system in protecting from severe disease upon re-exposure,...” This is speculation. If the SARS-CoV-2 test (PCR?) was negative, how does this then highlight the role of the immune system for protection of disease? Apparently the reason for the test is unknown (contact tracing vs. symptomatic episode), so this conclusion is not possible. Also, this fact that the participants got retested with a negative result does not demonstrate that “reinfections are more frequent than reported”.

We have now rewritten this section as follows: “Of note, about half of cluster participants underwent SARS-CoV-2 testing between three and six months, suggestive of exposure events or symptomatic episodes. None, however, reported testing positive. It is also possible that, rather than re-expansion, the decrease at one to three months reflects trafficking of lymphocytes from the circulation into the tissues. Indeed, levels of total CD4⁺ T cells, B cells and NK cells also tended to be lower in this cluster at these timepoints (Supplementary Fig. 4b). Studies of lung lavage and tissue samples from COVID-19 patients have demonstrated expression of tissue homing and tissue residence marker expression on T cells (Grau-Expósito et al., Nat Commun 2021; Saris et al., Thorax 2021; Liao et al., Nat Med 2021; Poon et al., Sci Immun 2021). As we were only able to obtain blood from participants, we were unable to address the issue of site-specific immunity in the current study, though a better understanding of the spatial nature of the immune response to infection is certainly necessary.”

REVIEWERS' COMMENTS

Reviewer #1 (Remarks to the Author):

The authors have responded in full to my previous comments.

Reviewer #2 (Remarks to the Author):

The authors have appropriately addressed my concerns. The additional information and analyses strengthen the manuscript and increase its clarity and value to readers.

Reviewer #3 (Remarks to the Author):

The authors could substantially improve the manuscript in this revision. In particular, broadening the experimental readouts with additional serological assays and a cell-free neutralization assay added significant value to the work. The authors can now better quantify waning of the immune response following SARS-CoV-2 infection. My comments have been sufficiently addressed.

Point-by-Point Response to Reviewers' Comments

Reviewer #1:

The authors have responded in full to my previous comments.

Reviewer #2:

The authors have appropriately addressed my concerns. The additional information and analyses strengthen the manuscript and increase its clarity and value to readers.

Reviewer #3:

The authors could substantially improve the manuscript in this revision. In particular, broadening the experimental readouts with additional serological assays and a cell-free neutralization assay added significant value to the work. The authors can now better quantify waning of the immune response following SARS-CoV-2 infection. My comments have been sufficiently addressed.

Author's Reply

We thank the three reviewers once again for their valuable time and their insightful comments. We are happy that the reviewers share our opinion that the revision and the additional data have substantially increased the depth and value of our work. There were no comments left to be addressed.